# Mitigating Reward Hacking in Inference-Time Alignment of Diffusion Models via Distributional Regularization

## Abstract

Diffusion models excel at generating images conditioned on text prompts, but the resulting images often do not satisfy user-specific criteria measured by scalar rewards such as Aesthetic Scores. This alignment typically requires fine-tuning, which is computationally demanding. Recently, inference-time alignment via noise optimization has emerged as an efficient alternative, modifying initial input noise to steer the diffusion denoising process towards generating high-reward images. However, this approach suffers from reward hacking, where the model produces images that score highly, yet deviate significantly from the original prompt. We show that noise-space regularization is insufficient and that preventing reward hacking requires an explicit image-space constraint. To this end, we propose **MIRA** (**MI**tigating **R**eward h**A**cking), a training-free, inference-time alignment method. MIRA introduces an image-space, score-based KL surrogate that regularizes the sampling trajectory with a frozen backbone, constraining the output distribution so reward can increase without off-distribution drift (reward hacking). Across SDv1.5 and SDXL, multiple rewards (Aesthetic, HPSv2, PickScore), and multiple public datasets (e.g., Animal-Animal, HPDv2), MIRA achieves $> 60\%$ win rate vs. strong baselines while preserving prompt adherence; mechanism plots show reward gains with near-zero drift, whereas DNO drifts as compute increases. We further introduce **MIRA-DPO**, mapping preference optimization to inference time with a frozen backbone, extending MIRA to non-differentiable rewards without fine-tuning.

## 1 Introduction

Text-to-image (T2I) diffusion models excel in generating high-fidelity images from textual prompts, yet aligning them to specific human preferences remains challenging (Liu et al., 2024). Inference-time alignment has emerged as a practical alternative to fine-tuning, enabling model alignment without retraining billions of parameters. Although inference-time alignment is well studied in LLMs (Li et al., 2025; Chakraborty et al., 2024), the diffusion setting is still evolving. A promising direction is noise optimization (Tang et al., 2024; Guo et al., 2024), which adjusts the injected starting noise to steer the denoising sampling path toward desired outcomes.

**Why noise optimization for inference-time alignment?** Other inference-time alignment methods, such as Best-of-$N$ (Nakano et al., 2021) and controlled denoising (Singh et al., 2025), generate multiple candidates and select the one with highest reward. Akin to an unguided search, they rely on the chance that a high-reward solution exists within a random batch of samples. If the chance of generating a high-reward image is low, these approaches are highly inefficient. In contrast, noise optimization is like a guided search process (see Figure 1). It uses the reward gradient to navigate in the initial noise space, steering the generation process toward more desirable images. By treating noise as a parameter to be optimized, it can generate rare high-reward images that Best-of-$N$ would likely miss. This approach provides a more powerful mechanism for alignment, but its increased flexibility to optimize rewards also introduces the challenge of *reward hacking*.

**The problem of reward hacking.** Noise optimization can exploit flaws in the reward model and produce high-scoring images that ignore the user's prompt. This is a symptom of reward hacking. For example, in Figure 2, we prompt the model with *"generate an image of a spider."* Although

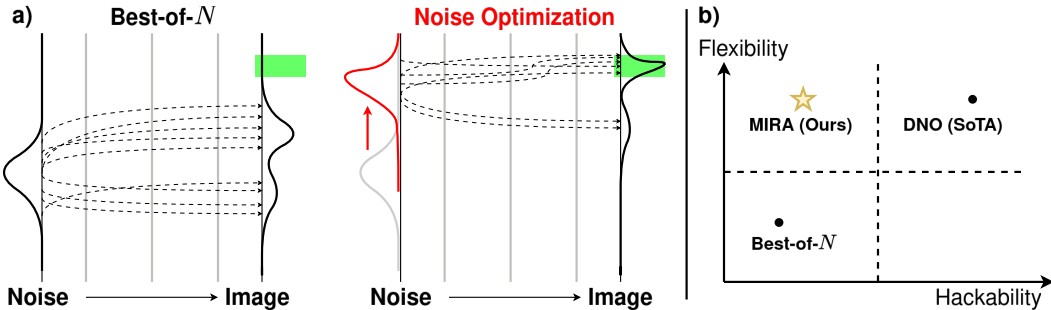

Figure 1: **Sampling-based methods vs noise optimization.** (a) Best-of-$N$ draws many independent samples from the base model and picks the best. When the green (high-reward region) in the image space has low probability density, most samples fall elsewhere and Best-of-$N$ is not efficient. Noise optimization, in contrast, shifts the noise distribution so generation is steered toward the high-reward region without large $N$. (b) Trade-off: Best-of-$N$ has lower hackability (semantic drift) but also lower flexibility (ability to steer outputs). Whereas DNO has higher flexibility but is prone to hacking, our method, MIRA, maintains flexibility and greatly reduces hackability.

the state-of-the-art DNO (Tang et al., 2024) generates an image with high reward (Aesthetic Score), the image does not depict a spider. By design, noise optimization pushes the diffusion model to generate images it normally would not produce, altering the output distribution. Reward models are trained to evaluate only the original types of images. On these new, out-of-distribution samples, their judgment becomes unreliable and their flaws can be exploited; this leads to reward hacking.

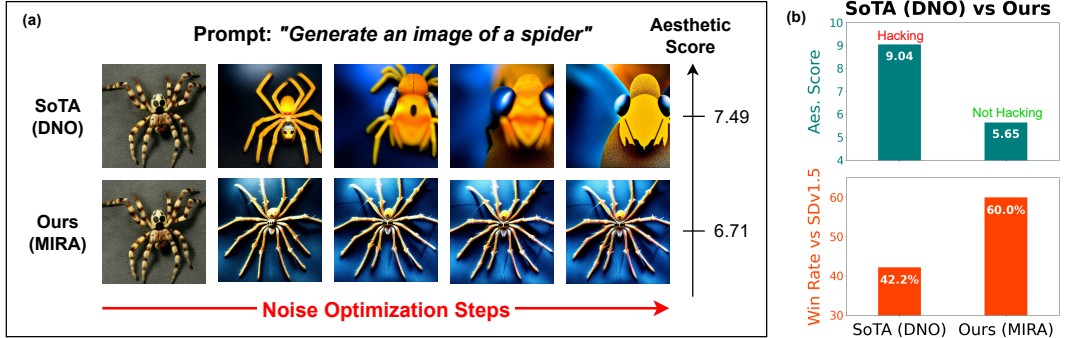

Figure 2: **Illustrating reward hacking in inference-time alignment of diffusion models.**
(a) Given the prompt *"generate an image of a spider"* and a preference for better aesthetics (Aesthetic Score (Schuhmann et al., 2022)), we observe the state-of-the-art (Direct Noise Optimization) achieves high reward yet no longer follows the prompt, an example of reward hacking. In contrast, our images have better aesthetic quality and do not suffer from reward hacking; hence, our reward is lower. (b) We obtain Aesthetic Score and win rate results (against base SDv1.5) on the Animal dataset (Black et al., 2023). We remark that MIRA (bottom) significantly outperforms the SoTA (top) in average win-rate, effectively mitigating reward hacking despite lower average rewards. We use GPT-4o (Hurst et al., 2024) as the win-rate judge.

**From key insight to MIRA.** While previous work (Tang et al., 2024) limits the size of noise updates, our analysis (Fig. 3, Prop. 1) shows that this is not enough; even small changes to the input noise can cause large, unpredictable changes in the final image. Our key insight, therefore, is that we must directly constrain the output distribution, keeping it similar to what the base model would normally produce. To this end, we propose **MIRA** (**MI**tigating **R**eward h**A**cking), a method that implements this control using a novel and efficient upper-bound surrogate for KL divergence, derived from the model's score functions. This penalty discourages large deviations between the images produced from the original noise ($z_0$) and the optimized noise ($z$), thereby preserving prompt adherence. To extend this framework to the common challenge of non-differentiable rewards, we also introduce MIRA-DPO, which optimizes directly from preference feedback to handle black-box objectives.

We summarize our main contributions as follows:

(1) A practical image-space regularizer for inference-time noise optimization, derived from a principled, score-based surrogate for KL divergence.

(2) A mechanism analysis demonstrating that reward hacking corresponds to distributional drift, quantitatively validated using both our surrogate and CMMD.

(3) An inference-time extension to pairwise preferences, MIRA-DPO, that adapts the DPO framework to frozen-backbone inference-time noise optimization.

On SDv1.5 and SDXL, across rewards (Aesthetic Score, Brightness, Darkness) and preference sets (Simple Animals, Animal–Animal, Animal–Object, HPDv2), MIRA achieves head-to-head win rates typically $> 60\%$ while reducing reward-hacking artifacts.

## 2    RELATED WORK

**Diffusion model alignment via fine-tuning.** Fine-tuning is a common alignment strategy, with methods employing techniques like reinforcement learning, preference optimization, and regression-based objectives (Prabhudesai et al., 2023; Clark et al., 2024; Black et al., 2023; Fan et al., 2024; Deng et al., 2024; Wu et al., 2024b; Li et al., 2024b; Wallace et al., 2024; Yang et al., 2024; Lee et al., 2023; Gu et al., 2024; Yuan et al., 2024). For a comprehensive survey, we refer the reader to (Uehara et al., 2024). However, the high computational cost of these approaches motivates the need for more efficient inference-time methods.

**Inference-time alignment methods.** More flexible and computationally efficient are inference-time alignment techniques. These include sampling methods like Best-of-$N$ (Nakano et al., 2021), various forms of guidance (Dhariwal & Nichol, 2021; Song et al., 2021b; Wu et al., 2024a; Dou & Song, 2024; Cardoso et al., 2023; Phillips et al., 2024; Li et al., 2024a; Naesseth et al., 2019; Yu et al., 2023), and trajectory steering methods (Song et al., 2023; Uehara et al., 2025b;a; Singhal et al., 2025; Kim et al., 2025; Singh et al., 2025). An orthogonal and promising direction is **noise optimization**. While amortized approaches like Golden Noise (Zhou et al., 2025) learn a static noise mapping offline, our work focuses on instance-specific optimization to allow plug-and-play alignment without prior training. Previous work in this instance-specific category has focused on improving prompt fidelity (using cross-attention maps (Guo et al., 2024) or auxiliary knowledge graphs (Xie & Gong, 2024)) or has proposed noise-space regularization to counteract reward hacking (Tang et al., 2024; Eyring et al., 2024). Despite these advances, reward hacking remains a notable challenge for existing noise optimization methods, which we aim to address in this work.

## 3    PROBLEM FORMULATION

A frozen conditional diffusion model $p_\theta(\cdot|c)$ generates an image by progressively denoising an initial noise sample $x_T \sim \mathcal{N}(0, \mathbf{I})$ over $T$ steps to produce an image $x_0$. For a given prompt $c$ and noise $z$, the goal of inference-time noise optimization is to minimize:

$$\mathcal{L}(z, c) := -\mathbb{E}_{x_0 \sim p_\theta(\cdot|z,c)}\big[r(x_0, c)\big], \tag{1}$$

where $r(\cdot, c)$ is a reward function. Equation 1 is solved with gradient descent:

$$z_{k+1} \leftarrow z_k - \alpha \nabla_z \mathcal{L}(z_k, c), \quad \text{learning rate } \alpha > 0, \tag{2}$$

with initial ($t = T$) noise $z_0 \sim \mathcal{N}(0, \mathbf{I})$. However, this simple reward maximization approach suffers from reward hacking (Chen et al., 2024; Miao et al., 2024a), often generating high contrast, overly saturated, or unnatural images that score highly while appearing unrealistic (as shown in Figure 2).

**Noise-space regularization is insufficient.** Existing methods address reward hacking by regularizing the input noise vectors (Tang et al., 2024). However, this noise-space regularization is fundamentally insufficient. The highly non-linear denoising process means that negligible perturbations to the input noise ($\|z - z_0\|_2 \ll \epsilon$) can still produce substantially different images (Figure 3). We formalize this insight as follows:

**Proposition 1.** *Closeness in the noise space does not imply closeness in the diffusion-induced image distribution $p_\theta(\cdot|z, c)$.*

We prove Proposition 1 in Appendix A.9; the KL divergence between image distributions is not controlled by the distance between their input noises. The failure of noise-space constraints therefore necessitates a new approach: regularizing the image distribution directly.

**Generalizing the formulation.** For clarity, we present the problem as optimizing a single initial noise vector $z$. However, this framework can be naturally extended to optimize the entire sequence of noises injected at each step of the denoising process. In our experiments, we adopt this more general approach to ensure a fair and direct comparison with state-of-the-art baselines like DNO, which operate on the full noise trajectory. Our proposed solution, MIRA, is directly applicable to this general case.

Figure 3: Tiny changes to the initial noise can yield markedly different images under the same prompt.

## 4 PROPOSED APPROACH

**Our key idea.** As established in Proposition 1, reward hacking arises because noise optimization can cause the output image distribution $p_\theta(\cdot|z, c)$ to deviate significantly from the base model's unoptimized distribution $p_\theta(\cdot|z_0, c)$. To address this directly, we introduce a regularizer that penalizes this divergence. This leads to the following loss for our method, **MIRA**:

$$\mathcal{L}_{\mathrm{MIRA}}(z, c) := \underbrace{-r(x_0, c)}_{\text{Negative Reward } (-r)} + \underbrace{\beta d_{\mathrm{KL}}\left[p_\theta(x_0|z, c)\,\|\,p_\theta(x_0|z_0, c)\right]}_{\text{Image Distribution Regularizer}}, \tag{3}$$

where $\beta > 0$ is a hyperparameter controlling the regularization strength. The noise vector $z$ is then optimized by gradient descent: $z_{k+1} \leftarrow z_k - \alpha \nabla_z \mathcal{L}_{\mathrm{MIRA}}(z_k, c)$. However, a critical challenge is that the KL divergence term in eq. (3) is intractable for high-dimensional images.

**A practically feasible approach.** To create a practical algorithm, we instead optimize a tractable upper-bound surrogate derived from the model's score functions $s(\cdot)$. Our final objective is:

$$\mathcal{L}_{\mathrm{MIRA}}(z, c) = -r(x_0, c) + \beta \mathbb{E}\left[\sum_{t=0}^{T-1} \sigma_t^2 \|s(x_t|z_0, c)\|^2 - \sigma_t^2 \|s(x_t|z, c)\|^2\right], \tag{4}$$

where $\sigma_t$ denotes the variance schedule. This formulation effectively mitigates reward hacking while being computationally feasible. We derive eq. (4) in Appendix A.10 and outline the detailed procedure in Algorithm 1. We provide a high-level overview of our method's workflow in fig. 4.

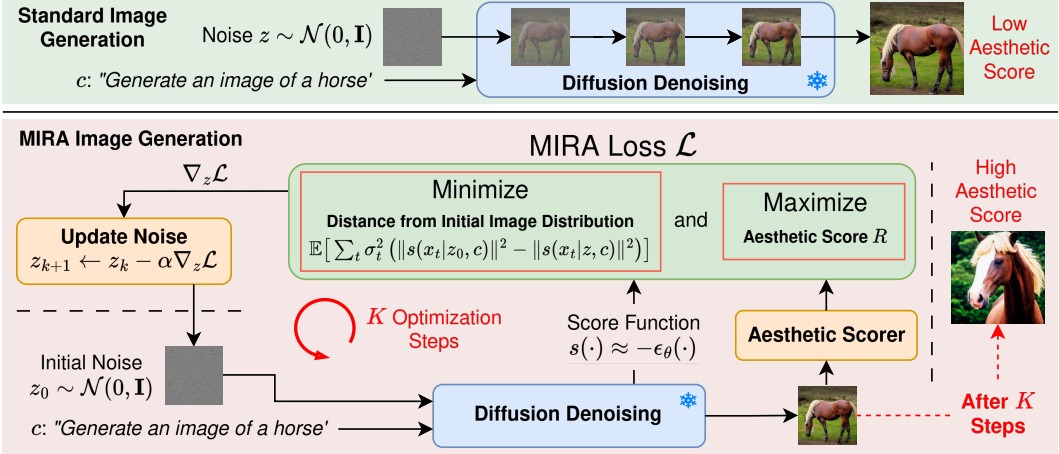

Figure 4: **Overview of our approach.** The green block illustrates the standard diffusion process in which a frozen diffusion model generates an image from an initial Gaussian noise. The red block represents our enhancement: we evaluate the generated image according to some metric (Aesthetic Score) and update the noise vector to minimize the MIRA loss. Our loss introduces an image-space regularization term, ensuring the sampling trajectory remains close to the model's original image distribution and mitigating semantic drift. By iteratively applying this process over $K$ steps, we achieve effective inference-time alignment through noise optimization.

---

**Algorithm 1** Proposed Algorithm: MIRA (for differentiable rewards)

---

**Require:** Initial noise $z_0 \sim \mathcal{N}(0, \mathbf{I})$, prompt $c$, reward function $r$, sampling process (e.g., DDIM) $G_\theta$, optimization steps $K_{\text{opt}}$, learning rate $\alpha$, regularization hyperparameter $\beta > 0$

1: Initialize $z \leftarrow z_0$
2: $\_, S_0 \leftarrow G_\theta(z_0, c)$        *// Obtain $S_0 = \sum_t \sigma_t^2 \|s(x_t|z_0, c)\|^2$ (eq. (4))*
3: **for** $k = 1, \ldots, K_{\text{opt}}$ **do**
4:    $x_0, S \leftarrow G_\theta(z, c)$        *// Generate $x_0$; obtain $S = \sum_t \sigma_t^2 \|s(x_t|z, c)\|^2$ (eq. (4))*
5:    $\mathcal{L}_{\text{MIRA}} \leftarrow -r(x_0, c) + \beta(S_0 - S)$        *// Compute loss*
6:    $z \leftarrow z - \alpha \nabla_z \mathcal{L}_{\text{MIRA}}$        *// Update noise*
7: **end for**
8: **return** $z$

---

### 4.1 A Novel Extension to Non-differentiable Rewards

MIRA is highly effective for aligning models with differentiable rewards. We now extend its core principles to handle non-differentiable objectives which include many practical and challenging alignment signals. For example, objectives such as JPEG compressibility (Black et al., 2023), Attribute Binding Score (Jia et al., 2024), and Recall Reward (Miao et al., 2024b) are often black-box and cannot be optimized with gradient-based methods.

To address this, we introduce **MIRA-DPO**, which adapts the Direct Preference Optimization (DPO) (Rafailov et al., 2024) framework to our entirely inference-time, frozen-backbone setting. This approach learns from preferences $(x_0^w, x_0^l)$, where $x_0^w$ is preferred over $x_0^l$. The DPO framework defines the probability of a preference using the Bradley-Terry model, which depends on a ground-truth reward function $r^*$. Framing this as a binary classification problem yields the negative log-likelihood loss:

$$\mathcal{L}_{\text{MIRA-DPO}} = -\mathbb{E}\Big[ \log \sigma \Big( r^*(x_0^w, c) - r^*(x_0^l, c) \Big) \Big]. \tag{5}$$

**The crucial insight** that connects this to our work is the relationship between the optimal reward $r^*$ and the diffusion model's probabilities. As shown in our full derivation (Appendix A.11), this reward is equivalent to the regularized log-likelihood ratio:

$$r^*(x_0, c) \propto \mathbb{E}\left[ \log \frac{p_\theta(x_{0:T}|z, c)}{p_\theta(x_{0:T}|z_0, c)} \right]. \tag{6}$$

**This term is exactly the quantity MIRA seeks to regularize.** However, this formulation presents a familiar bottleneck; the log-likelihood ratio is intractable for diffusion models. To create a practical algorithm, we thus approximate this term using our score-based KL surrogate, approximating $r^*$ as:

$$r^*(x_0, c) \approx \sum_{t=0}^{T-1} \sigma_t^2 \left( \|s(x_t|z_0, c)\|^2 - \|s(x_t|z, c)\|^2 \right), \tag{7}$$

which we use to optimize the loss in eq. (5). This allows MIRA-DPO to align with black-box objectives at inference time while still constraining semantic drift. We provide the full derivation in Appendix A.11 and the practical algorithm in Appendix A.2.

## 5 Experiments

We evaluate three questions central to our method's practicality:

- **Q1:** To what extent does MIRA mitigate reward hacking in noise optimization?
- **Q2:** How competitive is MIRA's alignment with state-of-the-art noise optimization baselines?
- **Q3:** How well can MIRA-DPO handle non-differentiable / black-box rewards?

In the following sections, we demonstrate that MIRA successfully mitigates reward hacking artifacts while improving prompt fidelity (Q1). We show that this leads to a superior alignment with human

preferences, achieving highly competitive win rates against state-of-the-art baselines (Q2). Finally, we establish that our MIRA-DPO framework robustly handles challenging non-differentiable objectives where gradient-based methods fail (Q3).

**Experimental Setup: Models, Datasets, and Evaluation.** We conduct experiments using two primary base models: Stable Diffusion v1.5 (SDv1.5) (Rombach et al., 2022) and SDXL (Podell et al., 2023). For each sample, we perform 50 optimization iterations. Within each iteration, we use DDIM sampling (Song et al., 2021a) with $\eta = 1$, a guidance scale of 5, and 100 sampling steps. We use this DDIM configuration for all backbones except SDXL-Turbo, for which we use the default sampler with one step. We further restrict our main scope to **noise optimization** baselines; sampling approaches (e.g., FK Steering, CoDe) are a promising and orthogonal line of work. Further implementation details can be found in Appendix A.1.

To test generalization, we report results on the Animal dataset from DDPO (Black et al., 2023), Animal-Animal and Animal-Object datasets from InitNO (Guo et al., 2024), and more complex prompts from HPDv2 (Wu et al., 2023). We use the format: *"generate an image of a* [PROMPT]*,"* where [PROMPT] is drawn from the Animal dataset (Black et al., 2023) for SDv1.5 and HPDv2 (Wu et al., 2023) for SDXL. For the reader's reference, we include all Animal prompts in Appendix A.7.

We evaluate performance using head-to-head win rate, a standard metric in image generation (Kirstain et al., 2023) measuring the percentage of prompts for which one method is preferred over another. We choose this as our primary metric because reward hacking can inflate raw scores, producing high-reward images that diverge from the prompt. Hence, while rewards alone are insufficient to reliably evaluate true model performance, win rates more directly reflect human preferences. We compare our method with several key inference-time noise optimization baselines: DNO (Tang et al., 2024), DyMO (Xie & Gong, 2024), InitNO (Guo et al., 2024), and ReNO (Eyring et al., 2024). We include reward values and CLIPScores in Appendix A.4, Tables 2 and 3.

## 5.1 To what extent does MIRA mitigate reward hacking in noise optimization?

In this section, we investigate whether MIRA can effectively mitigate reward hacking during inference-time alignment of diffusion models, while maintaining high prompt fidelity. To this end, we evaluate our approach against the state-of-the-art DNO (Tang et al., 2024) using a range of human-aligned reward models (Aesthetic Score (Schuhmann et al., 2022), HPSv2 (Wu et al., 2023), PickScore (Kirstain et al., 2023)), and custom image brightness and darkness rewards. The latter serve as stress tests to clearly visualize reward hacking behavior.

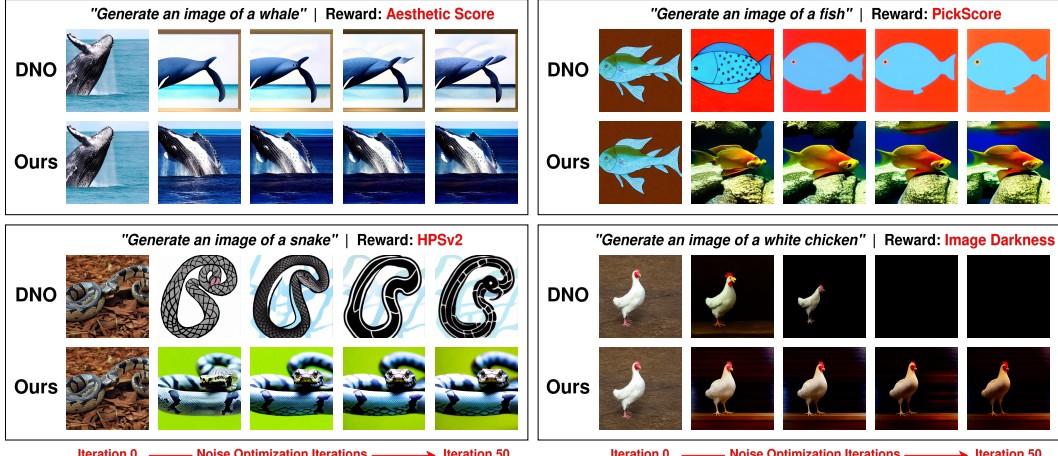

Figure 5: **MIRA reduces reward hacking on four rewards (Aesthetic Score, PickScore, HPSv2, Darkness).** For each prompt, we show seed-matched images over 50 noise optimization iterations for DNO (top row) and MIRA (bottom row). MIRA increases the target reward while preserving prompt adherence and texture realism; DNO drifts toward oversaturation artifacts and loss of detail. MIRA also improves over base SDv1.5 images (iteration 0), enhancing color richness.

Our experiments reveal a key observation. Although the state-of-the-art DNO (Tang et al., 2024) shows some effectiveness in aligning images with human-aligned reward functions, it exhibits clear

patterns of failure cases, particularly in maintaining prompt fidelity. Despite noise regularization, DNO can produce images with unnatural textures and lighting, leading to deviation from the prompt. In contrast, MIRA achieves a better balance between reward optimization and semantic alignment. As shown in Figure 5, MIRA produces visually coherent and accurate images across Aesthetic Score, HPSv2, PickScore, and a custom image darkness reward. **Optimizing for image darkness most clearly illustrates reward hacking;** DNO generates almost completely black outputs that technically score highly but ignore prompt details (e.g., omitting the "white cat"). MIRA mitigates this kind of reward hacking, balancing high reward scores and alignment with key visual concepts. We observe a similar pattern under the image brightness reward, with additional examples in Appendix A.3; our findings collectively reinforce that MIRA reduces reward hacking.

To understand the mechanism behind these qualitative results, we analyze the distributional drift during the optimization process. We measure this drift by computing a Monte Carlo approximation of our score-based KL surrogate, averaged over all 45 prompts in the Simple Animals dataset over five seeds. Due to the finite sample size, this estimate may exhibit sampling variance which can result in small negative values when the true distributional drift is near zero.

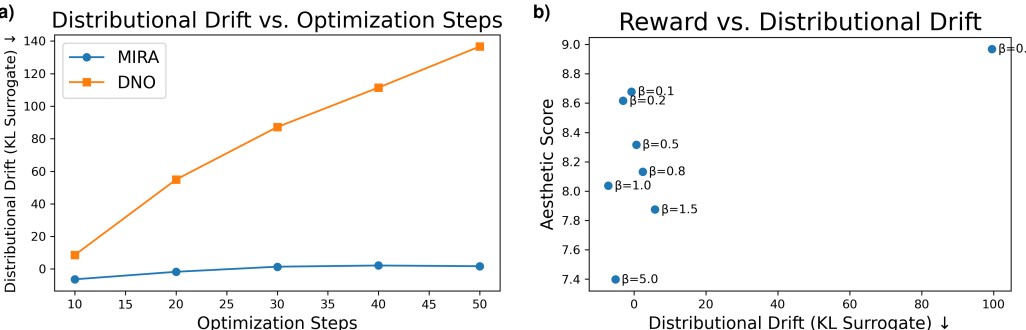

Figure 6: **Mechanism of MIRA's effectiveness, averaged over the Simple Animals dataset.** We plot our score-based KL surrogate as a proxy for distributional drift. **(a)** During optimization, MIRA's drift remains near-zero while DNO's grows significantly. **(b)** Sweeping the hyperparameter $\beta$ for MIRA reveals a stable trade-off, where a moderate value achieves a significant reward increase with negligible drift.

Our findings in Figure 6a show that DNO's average drift grows substantially with more optimization steps, indicating that it consistently achieves high rewards by producing out-of-distribution images. In contrast, MIRA's average drift remains near zero. Furthermore, Figure 6b illustrates the stable reward-drift trade-off. This core finding is validated by the CMMD (Jayasumana et al., 2024) metric measuring distributional distance. Comparing against a reference batch of 900 images (20 for each prompt in the Simple Animals dataset) generated by base SDv1.5, CMMD strongly supports our analysis. DNO exhibits a large distributional drift (CMMD of 0.281) while MIRA remains remarkably close to the base distribution (CMMD of 0.063, lower is better). Taken together, these results provide a clear explanation for our method's success: MIRA effectively increases rewards by finding better images that remain faithful to the original image distribution, thus avoiding the semantic degradation characteristic of reward hacking.

### 5.2 How competitive is MIRA's alignment with state-of-the-art noise optimization baselines?

Although reward metrics such as Aesthetic Score are commonly used to evaluate alignment, our results in Section 5.1 provide evidence that reward values cannot reliably measure model performance. We instead use pairwise win rate comparisons, using both GPT-4o Hurst et al. (2024) and human feedback to choose between two images generated from the same prompt. The GPT-4o evaluation prompt (Appendix A.8) balances prompt adherence, visual realism, and overall reward maximization.

Table 1 summarizes our results: MIRA demonstrates strong generalization and reduced reward hacking, consistently outperforming all baselines across datasets. Even on challenging multi-subject prompts (Animal-Animal and Animal-Object), it achieves win rates exceeding 50%. In direct com-

| Comparison | Animal | | | Animal-Animal | | | Animal-Object | | |
|---|---|---|---|---|---|---|---|---|---|
| | Aesthetic ↑ | Bright. ↑ | Dark. ↑ | Aesthetic ↑ | Bright. ↑ | Dark. ↑ | Aesthetic ↑ | Bright. ↑ | Dark. ↑ |
| MIRA vs DDPO (Black et al., 2023) | 60.00±5.12 | 73.33±5.21 | 83.33±2.90 | 57.58±6.00 | 60.61±5.96 | 83.33±4.27 | 54.86±6.00 | 68.06±7.10 | 81.25±5.53 |
| MIRA vs Diff-DPO (Wallace et al., 2024) | 62.22±4.97 | 91.11±1.99 | 93.33±1.22 | 66.67±6.59 | 81.82±5.12 | 84.85±5.31 | 63.19±6.54 | 82.64±3.65 | 89.58±1.86 |
| MIRA vs D3PO (Yang et al., 2024) | 73.33±4.61 | 87.64±3.72 | 84.44±6.55 | 63.64±6.29 | 68.18±3.65 | 74.24±5.35 | 68.06±5.35 | 77.78±5.31 | 83.33±5.79 |
| MIRA vs BoN (N=50) (Nakano et al., 2021) | 62.00±3.65 | 86.66±3.30 | 90.33±3.85 | 62.12±2.43 | 71.21±5.75 | 92.42±2.22 | 54.86±9.69 | 83.33±3.14 | 93.06±1.22 |
| MIRA vs InitNO (Guo et al., 2024) | 61.11±3.37 | 72.22±5.75 | 82.22±2.90 | 65.15±3.98 | 78.79±5.07 | 83.33±3.30 | 57.64±6.40 | 84.03±5.35 | 80.56±5.75 |
| MIRA vs DyMO (Xie & Gong, 2024) | 53.89±8.52 | 68.89±6.78 | 68.89±5.12 | 56.06±5.79 | 80.30±4.71 | 87.98±4.44 | 46.53±7.47 | 84.81±5.12 | 87.50±3.98 |
| MIRA vs DNO (Tang et al., 2024) | 80.00±2.22 | 77.78±6.74 | 64.44±7.13 | 77.27±4.18 | 57.58±5.12 | 66.67±5.79 | 68.06±1.86 | 77.78±5.96 | 71.53±5.58 |

Table 1: **GPT-4o win rates (%) of MIRA against baselines across diverse datasets and objectives.** The table details head-to-head comparisons on the Animal, Animal-Animal, and Animal-Object datasets for three reward functions (Aesthetic Score, Brightness, Darkness). On nearly all settings, MIRA achieves win rates significantly above 50%, showing a consistent preference over a wide range of state-of-the-art methods. Values indicate mean and standard error over five seeds. Reward models are highlighted in yellow.

parisons with base SDv1.5, our win rates reach 90% (see Appendix A.4 Table 1 and Figure 3), grounding our method's performance as substantial improvement over the unaligned base model.

For direct human feedback, we conducted a focused user study with 100 anonymous participants to collect direct human feedback on SDv1.5 images optimized for Aesthetic Score using MIRA and DNO. The ordering of image pairs is randomized, and additional study setup details are provided in Appendix A.1. As shown in Figure 7, participants prefer MIRA over DNO in more than 80% of comparisons, providing evidence of stronger alignment with human intent.

Finally, to test scalability, we extend MIRA to SDXL (Podell et al., 2023) and SDXL-Turbo (Sauer et al., 2024) backbones using the first 50 HPDv2 (Wu et al., 2023) prompts, optimizing for HPSv2. In separate user studies, we compare our method against leading alternatives: DNO on SDXL and ReNO (Eyring et al., 2024) on SDXL-Turbo. As shown in Figure 7, the results demonstrate a clear preference for our approach. MIRA achieves a 55.75% win rate over DNO and a 63.35% win rate over ReNO. This win rate against ReNO is supported by qualitative examples (Appendix A.3 Figure 2), where MIRA tends to produce more literal and detailed compositions on the SDXL-Turbo backbone.

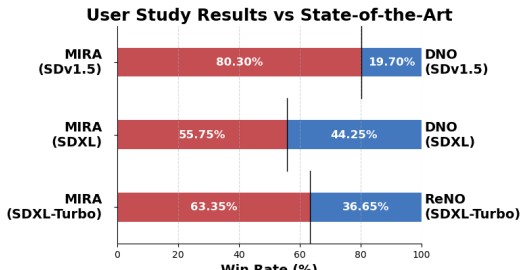

Figure 7: **Human preference win rates for MIRA vs. state-of-the-art baselines.** The chart displays head-to-head win rates from three focused user studies. **Top:** On SDv1.5 (optimizing Animal prompts for Aesthetic Score), MIRA achieves an 80.30% win rate against DNO. **Middle and Bottom:** On SDXL and SDXL-Turbo (optimizing HPDv2 subset for HPSv2), MIRA achieves win rates of 55.75% against DNO and 63.35% against ReNO, respectively.

In summary, the evidence presented in this section provides a comprehensive answer to our second research question. Through large-scale automated evaluations, MIRA consistently outperforms a wide range of both fine-tuning and inference-time baselines. This strong quantitative performance is corroborated by focused human studies, where MIRA is preferred over the state-of-the-art DNO by a 4-to-1 margin on SDv1.5 and also wins decisively on the more powerful SDXL and SDXL-Turbo backbones. Together, these findings establish MIRA as a highly competitive method for inference-time alignment.

### 5.3 HOW WELL CAN MIRA-DPO HANDLE NON-DIFFERENTIABLE/BLACK-BOX REWARDS?

Many real-world alignment objectives, such as human preferences or perceptual metrics, are inherently non-differentiable and cannot be directly optimized using gradient-based methods. We evaluate MIRA-DPO through both quantitative benchmarking and qualitative robustness tests.

**Quantitative Evaluation (Aesthetic Score).** We treat Aesthetic Score as a black-box reward. We optimize for 50 steps on the Simple Animals dataset with $\beta = 0.2$ and average over five seeds. As reported in Table 2, MIRA-DPO achieves a win rate of 63.56% over DNO. This validates that our score-based surrogate effectively guides optimization even when exact gradients are unavailable.

**Qualitative Evaluation (JPEG Compressibility).** We further evaluate on the task of JPEG compressibility (Reich et al., 2024; Wallace, 1991), inherently non-differentiable due to rounding oper-

| Comparison | Aesthetic Score Win Rate (%) ↑ |
|---|---|
| MIRA-DPO vs. DNO (Hybrid-2) | 63.56 ± 3.61 |

Table 2: **Quantitative validation of MIRA-DPO.** We compare MIRA-DPO against DNO's Hybrid-2 variant by treating Aesthetic Score as a non-differentiable reward. MIRA-DPO outperforms the baseline, demonstrating the effectiveness of the score-based surrogate in the black-box setting.

ations. This objective creates a strong incentive for reward hacking by removing textures to lower file size. As shown in Figure 8, the DNO baseline collapses the image into a nearly blank state to minimize file size. In contrast, MIRA-DPO compresses the image while preserving the semantic structure and core features of the prompt, demonstrating that our image-space regularization prevents semantic collapse even under adversarial objectives.

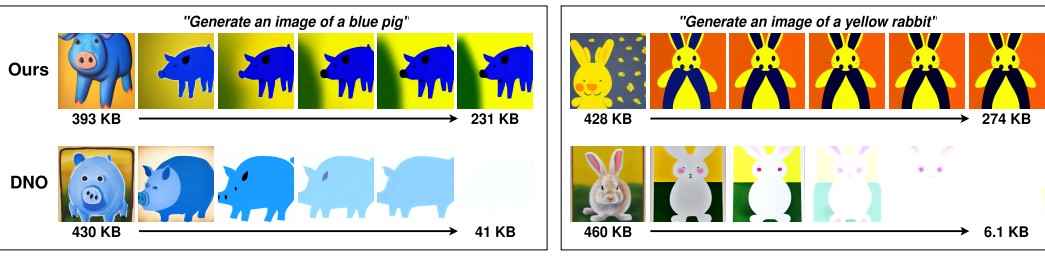

Figure 8: **MIRA-DPO preserves semantic content while optimizing a non-differentiable reward.** We compare MIRA-DPO with DNO on maximizing JPEG compressibility. We use prompts *"generate an image of a blue pig"* (left) and *"generate an image of a yellow rabbit"* (right). While DNO's output degrades into a nearly blank image to hack the file size metric, MIRA-DPO achieves compression while retaining semantic coherence.

## 5.4 ABLATIONS AND ADDITIONAL EXPERIMENTS

In addition to win rate results shown in the above sections, we report the mean reward values for all methods (including MIRA) across all Animal prompts in Appendix A.4 Table 2. These values provide important context for interpreting trade-offs between reward maximization and prompt adherence for each objective (Aesthetic Score, HPSv2, PickScore, brightness, and darkness). Furthermore, we tune the hyperparameter $\beta$ to balance the trade-off between reward maximization and win rate. As shown in Appendix A.6 Figure 6, increasing $\beta$ systematically reduces reward scores but improves CLIPScore (quantifying image-prompt alignment). These ablations reveal that low $\beta$ tends toward reward hacking, whereas high $\beta$ overconstrains, leading to an under-optimized reward.

**Efficiency Comparison with Sampling-Based Methods.** Next, we compare MIRA to steering / sampling-based methods such as FK Steering (Singhal et al., 2025) and CoDe (Singh et al., 2025), matching MIRA's runtime with those of each baseline. All sampling methods run with their default configurations (SDv1.5 backbone) on one A100. For each method, we fix a single configuration, measure its mean per-prompt runtime $T$ on the Simple Animals prompts, and then run MIRA for the same budget $T$ on the same prompts. We report our head-to-head win rates for Aesthetic Score, darkness, and brightness rewards. This serves as a stress test where sampling-based methods can be sample-inefficient.

| Comparison | Aesthetic Score Win Rate (%) ↑ | Darkness Win Rate (%) ↑ | Brightness Win Rate (%) ↑ | Peak GPU Mem (GB) ↓ |
|---|---|---|---|---|
| MIRA vs FK Steering (16 Particles) | 57.78 | 75.56 | 73.33 | 20.55 |
| MIRA vs DAS (8 Particles) | 37.78 | 77.78 | 80.00 | 23.90 |
| MIRA vs CoDe ($N=20$) | 55.56 | 73.33 | 77.78 | 24.51 |
| MIRA vs Best-of-$N$ ($N=50$) | 53.33 | 80.00 | 82.22 | 2.63 |

Table 3: **Comparison against state-of-the-art steering/sampling-based methods with matched wall-clock time budget.** For each baseline, the table lists peak GPU memory usage of each sampling-based method as well as MIRA's head-to-head win rates on Aesthetic Score, darkness, and brightness rewards. While MIRA's peak memory remains constant at 8.97GB, most steering-based competitors require over 20GB, highlighting a significant memory advantage for our method. Under a matched time budget, MIRA demonstrates highly competitive performance, particularly on brightness/darkness rewards where it consistently achieves win rates over 73%.

**Compositional Alignment on Diffusion Transformers.** Finally, to demonstrate the generalization of MIRA to Diffusion Transformer (DiT) architectures (Peebles & Xie, 2023) and validate performance on standardized benchmarks, we apply our method to the one-step PixArt-$\alpha$ DMD (Chen et al., 2023). We utilize the T2I-CompBench (Huang et al., 2023) suite to evaluate compositional alignment across attribute binding, object relationships, and complex prompts. We optimize a composite reward function (ImageReward, HPSv2, PickScore, CLIPScore) following the configuration in Eyring et al. (2024). As shown in Table 4, MIRA consistently improves over the base PixArt-$\alpha$ DMD model across all metrics. This demonstrates that the score-based objective effectively generalizes to DiT backbones and enhances compositional alignment.

Table 4: **Compositional Evaluation on PixArt-$\alpha$.** We evaluate MIRA on T2I-CompBench (Huang et al., 2023) using PixArt-$\alpha$ (Chen et al., 2023) DMD (diffusion model distillation), a DiT architecture. MIRA consistently improves over the base model across all metrics, validating the method's robustness on modern backbones.

| Model | Attribute Binding | | | Object Relationship | | Complex ↑ |
|---|---|---|---|---|---|---|
| | Color ↑ | Shape ↑ | Texture ↑ | Spatial ↑ | Non-Spatial ↑ | |
| PixArt-$\alpha$ DMD | 0.378 | 0.342 | 0.469 | 0.179 | 0.306 | 0.291 |
| PixArt-$\alpha$ DMD + MIRA (Ours) | **0.455** | **0.444** | **0.583** | **0.216** | **0.321** | **0.347** |

## 6 CONCLUSIONS AND LIMITATIONS

In this paper, we introduce MIRA, a distribution-regularized noise optimization method for inference-time alignment of diffusion models that mitigates reward hacking. By reformulating noise optimization as a constrained reward maximization problem, MIRA updates latent noise during sampling to improve reward while adhering to the user's prompt. Extensive experiments demonstrate MIRA's strong empirical performance, with head-to-head win rates typically exceeding 60% against baselines. Our work establishes that the key to mitigating such reward hacking is to directly regularize the output image distribution, which is more robust than prior noise-space constraints. Notably, this insight is independently validated by concurrent work (Li & He, 2025), which argues that high-dimensional noise quantities lack the manifold structure necessary for robust modeling. MIRA provides the first empirical realization of this principle in the context of inference-time alignment.

**Limitations.** MIRA adds moderate inference-time compute beyond normal DDIM/DDPM sampling and relies on principled score-based surrogates, which may degrade on complex or out-of-distribution prompts; a formal analysis of the surrogate's theoretical tightness is an important direction for future work. While our score-based objective is sampler-portable, we focus our evaluation on DDIM for compute parity as hyperparameters may vary by schedule. Improving MIRA's efficiency and combining it with sampling methods are left to future work.

## 7 ETHICS STATEMENT

This work introduces MIRA, a method for improving the alignment of text-to-image diffusion models at inference time. We have considered the ethical implications of our research in accordance with the ICLR Code of Ethics.

- **Human Subjects:** Our research involved focused user studies with 100 anonymous online participants to evaluate image preferences. The task was low-risk, involving viewing and selecting between safe AI-generated images based on personal aesthetic preference and prompt faithfulness. No personally identifiable information was collected, and the study was designed to be brief to respect participants' time.

- **Intended Use and Misuse:** The intended purpose of MIRA is to make generative models more reliable and faithful to user intent, which is a positive alignment goal. By mitigating reward hacking, our method reduces the likelihood of models producing bizarre or unintended artifacts. However, like all powerful generative technologies, the underlying models can be misused for creating harmful or misleading content. MIRA itself does not introduce new misuse capabilities

but rather improves the control of an existing technology. We advocate for the responsible use of generative models and the continued development of robust safety and alignment techniques.

- **Bias in Models and Data:** MIRA is an inference-time method that operates on pre-trained, frozen-backbone models (e.g., Stable Diffusion). As such, it does not retrain or modify the underlying model weights and will therefore inherit any social or demographic biases present in the original models and their training data. Our work does not address these inherent biases, which remains a critical and open area of research for the community.

## 8 REPRODUCIBILITY STATEMENT

To promote reproducibility, we provide detailed documentation throughout the paper. MIRA is training-free and runs at inference time. The core mathematical formulations for MIRA's KL surrogate and MIRA-DPO's preference optimization are derived in App. A.10 and App. A.11. We document all hyperparameters (optimization iterations, learning rate, sampler, DDIM steps, $\eta$, CFG, $\beta$) in §5 and App. A.1. We detail the exact GPT-4o evaluation prompt in App. A.8 and cite the appropriate datasets for our prompts. Additionally, we describe the user study protocols in App. A.1. We will release, upon acceptance, (i) source code, (ii) scripts to reproduce tables with fixed seeds. Baselines are cited and run with public configs; hardware and runtime details are in App. A.1.

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

# A APPENDIX

## A.1 IMPLEMENTATION DETAILS

We demonstrate results of MIRA on multiple reward functions, including Aesthetic Score, HPSv2, PickScore, and image brightness and darkness. The first three are human-aligned, whereas the last two are useful for easily visualizing reward hacking. We set $\beta = 0.2$ for Aesthetic Score, $\beta = 0.5$ for HPSv2 and PickScore, $\beta = 1$ for brightness, and $\beta = 0.8$ for darkness. For the single-step SDXL-Turbo experiments (HPSv2), we set $\beta = 0.05$. For JPEG compressibility, a non-differentiable reward, we use $\beta = 1$. HPSv2 and PickScore use ViT-H/14 as the backbone, and Aesthetic Score uses ViT-L/14.

We implement MIRA with a learning rate of $0.01$ and the AdamW optimizer in all experiments. Our method runs on a single A100 Nvidia GPU. For DNO, we use noise regularization (i.e. PRNO) unless otherwise stated. Other baselines use their default configurations.

**User Study Details.** We obtain these results from anonymous volunteers who respond using an online form. We ask these participants: *"Which of the below images would you personally prefer getting given the above prompt (based on your personal trade-off between prompt faithfulness and aesthetics)?"* We adopt this from ReNO. Given side-by-side images (prompt-matched, order-randomized), participants have the option to choose the left image, the right image, or neither/both. In the case of the last option, we assign a score of $0.5$ to that image pair. Due to computational restrictions, we limit the number of participants to 100. We further limit the number of images to 50 to respect the participants' time.

## A.2 ALGORITHMS

For the reader's reference, we provide our loss for non-differentiable rewards as follows. Given prompt $c$,

$$
\mathcal{L}_{\text{MIRA-DPO}} = -\mathbb{E}_{x_0^w, x_0^l} \log \sigma \left( \beta \mathbb{E}_{\substack{x_{1:T}^w \sim p_\theta(\cdot | x_0^w, z^w, c) \\ x_{1:T}^l \sim p_\theta(\cdot | x_0^l, z^l, c)}} \left[ \log \frac{p_\theta(x_{0:T}^w | z^w, c)}{p_\theta(x_{0:T}^w | z_0^w, c)} - \log \frac{p_\theta(x_{0:T}^l | z^l, c)}{p_\theta(x_{0:T}^l | z_0^l, c)} \right] \right).
$$

(8)

---

**Algorithm 2** MIRA for Non-differentiable Rewards

---

**Require:** Initial noise $z_0^{(1)}, z_0^{(2)} \sim \mathcal{N}(0, \mathbf{I})$, prompt $c$, reward function $r$, sampling process $G_\theta$, optimization steps $K_{\text{opt}}$, learning rate $\alpha$, regularization hyperparameter $\beta > 0$

1: Initialize $z^{(1)} \leftarrow z_0^{(1)}, z^{(2)} \leftarrow z_0^{(2)}$
2: **for** $k = 1, \ldots, K_{\text{opt}}$ **do**
3:     $x_0^{(1)} \leftarrow G_\theta(z^{(1)}, c)$             // *Generate images*
4:     $x_0^{(2)} \leftarrow G_\theta(z^{(2)}, c)$
5:     **if** $r(x_0^{(1)}, c) > r(x_0^{(2)}, c)$ **then**       // *Choose preferred image; assign winner and loser*
        $x_0^w \leftarrow x_0^{(1)}, x_0^l \leftarrow x_0^{(2)}$
        $z^w \leftarrow z^{(1)}, z^l \leftarrow z^{(2)}$
        $z_0^w \leftarrow z_0^{(1)}, z_0^l \leftarrow z_0^{(2)}$
6:     **else**
        $x_0^w \leftarrow x_0^{(2)}, x_0^l \leftarrow x_0^{(1)}$
        $z^w \leftarrow z^{(2)}, z^l \leftarrow z^{(1)}$
        $z_0^w \leftarrow z_0^{(2)}, z_0^l \leftarrow z_0^{(1)}$
7:     $\mathcal{L}_{\text{MIRA-DPO}} \leftarrow \text{compute\_loss}(x_0^w, x_0^l, z^w, z^l, z_0^w, z_0^l)$    // *Compute loss according to eq. (8)*
8:     $z^w \leftarrow z^w - \alpha \nabla_{z^w} \mathcal{L}_{\text{MIRA-DPO}}$      // *Update noise vectors (also updating $z^{(1)}$ and $z^{(2)}$)*
9:     $z^l \leftarrow z^l - \alpha \nabla_{z^l} \mathcal{L}_{\text{MIRA-DPO}}$
10: **end for**
11: **return** $z^w$

---

## A.3 ADDITIONAL QUALITATIVE RESULTS

**Brightness and darkness reward hacking in MIRA vs DNO**

Here, we show reward hacking on image brightness and darkness. We prompt the diffusion model with *"generate an image of a black* [ANIMAL]*"* or *"generate an image of a white* [ANIMAL]*"* respectively. The contrast between the prompt and the reward provides a clear visualization of reward hacking.

Figure 1: **MIRA vs DNO in reward hacking**. On the Image Brightness reward, we demonstrate that MIRA is able to effectively mitigate reward hacking and generate better, more realistic images while maintaining prompt fidelity, when compared to the state-of-the-art baseline on the same seed. In the top row, after 50 optimization steps, DNO completely hacks the brightness reward and generates an image that is overly white and unrealistic. In contrast, in the bottom row, MIRA is able to mitigate reward hacking and produce much better images, while aligning with the target reward.

**Qualitative results: MIRA vs ReNO on SDXL-Turbo**

In Figure 2, we compare our method (left) and ReNO (right) on SDXL-Turbo with identical prompts and an HPSv2 objective. Under equivalent sampling settings, MIRA tends to produce cleaner, more literal interpretations with stable composition and style consistency (e.g., large-scale scene layout, totem-like geometry, and concept blending), whereas ReNO often introduces extra flourishes or alternative readings. These examples math our quantitative trends that MIRA delivers stronger alignment than noise-only regularization.

Figure 2: **Qualitative comparison on SDXL-Turbo (left: MIRA, right: ReNO), optimized for HPSv2.** Prompts (top to bottom): "Wojak looking over a sea of memes from a cliff on 4chan," "Link fights an octorok in a cave in a Don Bluth style," "Totem pole made of cats," "Groot depicted as a flower." Across these cases, MIRA offers crisper compositions and more literal, prompt-faithful structure (e.g., coherent large-scene layout, pole-like stacking, integrated concept depiction), while ReNO reflects plausible but looser interpretations. Images are shown at equal inference settings for both methods.

## A.4 ADDITIONAL QUANTITATIVE RESULTS

In this section, we present additional results comparing MIRA to baselines.

**Win rates of all methods against SDv1.5**

In Table 1 below, we present the win rates of several methods including MIRA against SDv1.5 using Animal prompts. The results demonstrate that our method achieves the highest win rate in three rewards: Aesthetic Score, brightness, and darkness. We also represent this in the bar plot diagram in Figure 3. MIRA consistently outperforms baselines in win rates despite lower average rewards.

| Method (vs SDv1.5) | Aesthetic ↑ | Brightness ↑ | Darkness ↑ |
|---|---|---|---|
| DDPO | 51.11 | 80.00 | 71.11 |
| Diffusion-DPO | 48.89 | 77.77 | 66.67 |
| D3PO | 53.33 | 71.22 | 71.11 |
| BoN ($N = 30$) | 51.11 | 80.11 | 62.22 |
| BoN ($N = 40$) | 55.55 | 77.77 | 63.33 |
| BoN ($N = 50$) | 55.55 | 75.55 | 61.11 |
| DNO | 42.22 | 75.56 | 66.67 |
| Ours | **60.00** | **91.11** | **88.89** |

Table 1: **Win rates of all methods against SDv1.5.** The table reports the percentage of pairwise comparisons between all methods (including MIRA) vs SDv1.5 on three objectives: *Aesthetic Score, Brightness,* and *Darkness* with GPT-4o as the judge. We sample Best-of-$N$ (BoN) with base SDv1.5. Across all objectives, our method consistently achieves higher win rates than SDv1.5. These results demonstrate that our method not only outperforms previous baselines in terms of win rates over various target objectives, but is also able to generate better images while more effectively optimizing the target reward functions.

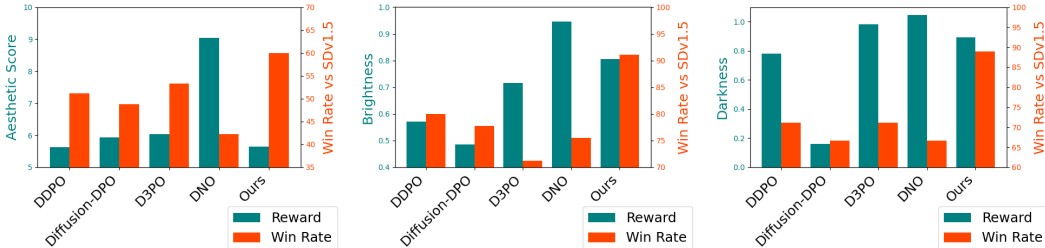

Figure 3: **Win rates (vs. SDv1.5) and average rewards across methods.** We evaluate our method, MIRA, on Aesthetic Score (left), brightness (middle), and darkness (right), comparing against DDPO, Diffusion-DPO, D3PO, and DNO. MIRA consistently outperforms baselines in win rates despite lower average rewards. Notably, higher rewards can indicate overoptimization, resulting in lower win rates.

**Overall rewards across methods and reward functions.** Table 2 presents the mean rewards of images generated by different methods (including MIRA). Rewards are averaged across all images generated from the Animal prompts.

| Method | Training Time | Inference Time | Aesthetic | HPSv2 | PickScore | Brightness | Darkness |
|---|---|---|---|---|---|---|---|
| SDv1.5 | - | 0.04 min | 5.367 | 0.278 | 20.390 | -0.123 | 0.087 |
| DDPO | 12 h | - | 5.623 | 0.307 | 20.120 | 0.570 | 0.782 |
| Diffusion-DPO | 2 h | - | 5.930 | 0.302 | 21.086 | 0.485 | 0.159 |
| D3PO | 24 h | - | 6.027 | 0.297 | 20.030 | 0.715 | 0.982 |
| BoN ($N = 30$) | - | 0.9 min | 6.044 | 0.318 | 21.404 | 0.006 | 0.329 |
| BoN ($N = 40$) | - | 1.31 min | 6.080 | 0.320 | 21.427 | 0.094 | 0.339 |
| BoN ($N = 50$) | - | 1.64 min | 6.111 | 0.321 | 21.365 | 0.105 | 0.353 |
| InitNO | - | 0.5 min | 5.259 | 0.286 | 20.120 | -0.041 | 0.113 |
| DyMO | - | 0.85 min | 5.720 | 0.322 | 21.220 | 0.320 | 0.008 |
| DNO | - | 5 min | 9.044 | 0.287 | 25.568 | 0.945 | 1.044 |
| Ours | - | 5 min | 5.646 | 0.285 | 21.161 | 0.805 | 0.894 |

Table 2: **Overall rewards across methods and reward functions.** The above table compares our method with other baselines across various reward metrics and runtime. These rewards include Aesthetic Score, HPSv2, PickScore, brightness, and darkness. The leftmost column contains all methods. The second column (runtime) lists the approximate time needed to fine-tune or optimize each method (for a single image). We note DDPO, Diffusion-DPO, and D3PO take several hours due to fine-tuning. Though our method does not generate images with the highest rewards, we observe higher rewards compared to SDv1.5 after just five minutes of optimization with less reward hacking as discussed in the main paper. Inference times are measured using a single A100 GPU.

**CLIPScores on the Simple Animals dataset.** We optimize several key methods (including fine-tuning methods) for Aesthetic Score on the Simple Animals dataset, evaluating CLIPScore as a prompt fidelity check. We observe that strong reward gains may come with weaker prompt adherence. Our method, MIRA, achieves the highest CLIPScore while remaining competitive on the actual target reward. We emphasize that *these results should not be considered in a vacuum, and win rate is a more holistic metric.*

| Method | CLIPScore | Aesthetic Score |
|---|---|---|
| DDPO | 24.898 | 5.623 |
| Diffusion-DPO | 25.340 | 5.930 |
| D3PO | 23.298 | 6.027 |
| BoN ($N = 50$) | 24.070 | 6.044 |
| InitNO | 24.555 | 5.720 |
| DyMO | 25.397 | 5.720 |
| DNO | 22.959 | 9.044 |
| MIRA (Ours) | 25.964 | 5.646 |

Table 3: **Comparison of CLIPScores according to method used, optimizing for Aesthetic Score.** We report CLIPScores, measuring prompt fidelity, and Aesthetic Score for various methods. MIRA achieves the best CLIPScore with comparable Aesthetic Score, whereas DNO achieves substantially higher Aesthetic Scores at the cost of prompt adherence.

**Hybrid feasibility (MIRA + FK Steering).** We implemented a simple hybrid combining MIRA with FK Steering to test complementarity. Concretely, we run multiple denoising trajectories in parallel and, at periodic intervals, select the candidate that maximizes the MIRA objective. The backbone remains frozen and configurations are fixed. This check is not wall-clock matched and serves only as a feasibility probe. On Simple Animals (45 prompts), optimizing for Aesthetic Score with $\beta = 0.1$, **MIRA with FK Steering achieves a 55.56% win rate over FK Steering alone (same backbone/GPU).**

**Trade-offs with compute.** We examine how reward changes as we vary the sampling budget while keeping the optimization procedure fixed. We sweep the number of DDIM sampling steps (NFE)

and compare MIRA to DNO under the same backbone and guidance. If the reward for a method increases significantly with additional sampling computation, this is indicative of reward hacking rather than genuine improvement. As shown in Fig. 4, DNO's reward increases drastically with more DDIM steps, whereas MIRA remains comparatively stable. This is consistent with MIRA's regularization that prevents the sampler from drifting toward high-score but unnatural solutions.

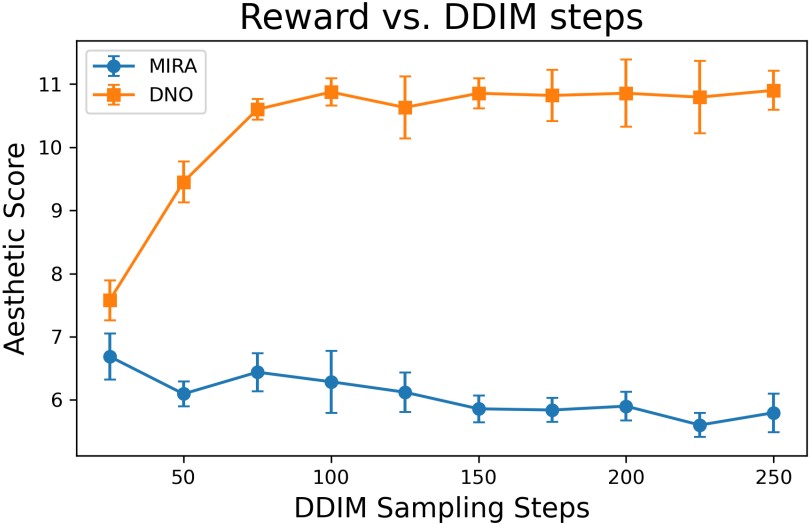

Figure 4: **Reward vs DDIM Steps (sampling compute) on Simple Animals dataset.** Mean Aesthetic Score vs. number of DDIM sampling steps ($\eta = 1$, same backbone/CFG; optimization iterations fixed). DNO's reward grows steadily with more steps, a hallmark of reward hacking; MIRA stays nearly flat (slight decline), indicating robustness to additional sampling compute. Error bars show variability across prompts. **The takeaway:** extra sampling compute disproportionately benefits noise-only optimization (DNO), while MIRA's image-space regularization curbs this effect.

### A.5    EMPIRICAL VALIDATION OF KL SURROGATE

To empirically validate the theoretical derivation of our score-based regularizer (Eq. 4), we conducted a controlled experiment in a low-dimensional setting, as calculating the true KL divergence in high-dimensional image spaces is intractable. We train a 1D UNet diffusion model (dimension 32, no downsampling) on a standard Gaussian $\mathcal{N}(0, \mathbf{I})$ for 3000 steps using Adam with a learning rate of $10^{-3}$. We then optimize the noise over 50 steps with a learning rate of $0.1$ to shift the generated distribution toward a target mean of $\mu = 2.5$. At each step, we computed the "True KL" (approximated by Kernel Density Estimation on 5000 samples) and our KL surrogate. We note that while Kernel Density Estimation (KDE) provides a near-exact estimate in this 1D setting, it is infeasible to apply in high-dimensional image manifolds due to the curse of dimensionality, necessitating the use of our score-based surrogate.

As shown in Figure 5, we observe a strict monotonic correlation between our MIRA KL surrogate (dashed blue) and the "True KL" (solid red). The difference in absolute scale is expected due to their mathematical definitions; the KL divergence integrates log-probability ratios ($\log \frac{p}{q}$) whereas our surrogate sums squared score vectors ($\|\nabla \log p\|^2$). Crucially, this strong correlation persists throughout the optimization process as the distribution changes. This confirms that minimizing our score-based surrogate effectively minimizes the true distributional divergence, providing a stable gradient signal regardless of the specific noise schedule encountered during inference.

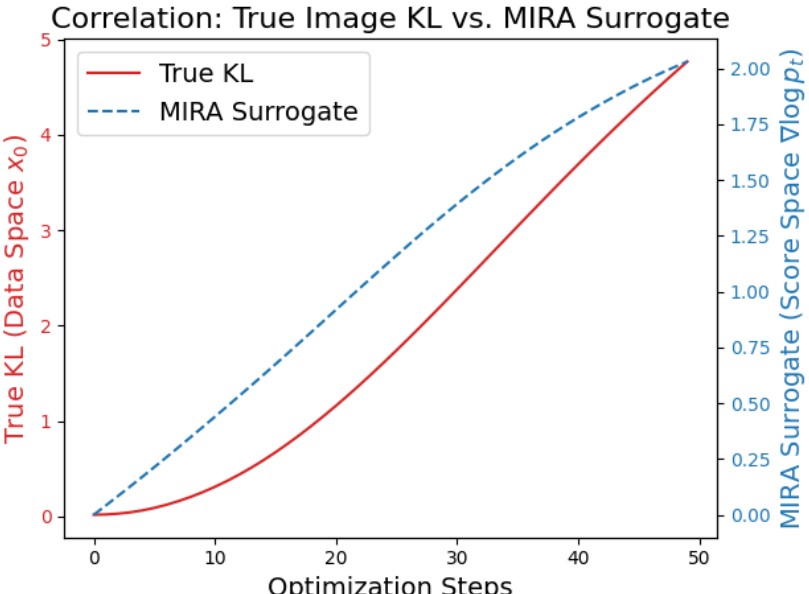

Figure 5: **Empirical validation of the MIRA surrogate on a 1D diffusion toy problem.** We compare the Data-Space "True" KL divergence (red, estimated via KDE) against our score-based MIRA surrogate (blue) over the course of 50 noise optimization iterations. The strong monotonic correlation confirms that our tractable score-based objective acts as a valid proxy for actual KL divergence even as the distribution deviates significantly from the prior.

## A.6 ABLATION ANALYSIS

**Impact of $\beta$ on CLIPScores and Rewards**

We justify our choice of hyperparameter $\beta$ by evaluating results across five different seeds, plotting the mean curves for CLIPScore and reward against optimization steps. CLIPScore is a metric which scores how well an image aligns with a given prompt. For our results, we use CLIPScore with a ViT-L/14 vision backbone and optimize for image darkness. As shown in Figure 6 increasing $\beta$ in our method leads to lower rewards but higher CLIPScores. We note that "early stopping" would yield higher CLIPScores but lower rewards.

## A.7 ANIMAL PROMPTS LIST

We use the following set of Animal DDPO prompts.

| cat | dog | horse | monkey | rabbit | zebra | spider | bird | sheep |
|------|--------|-------|-----------|--------|---------|---------|----------|----------|
| deer | cow | goat | lion | tiger | bear | raccoon | fox | wolf |
| lizard | beetle | ant | butterfly | fish | shark | whale | dolphin | squirrel |
| mouse | rat | snake | turtle | frog | chicken | duck | goose | bee |
| pig | turkey | fly | llama | camel | bat | gorilla | hedgehog | kangaroo |

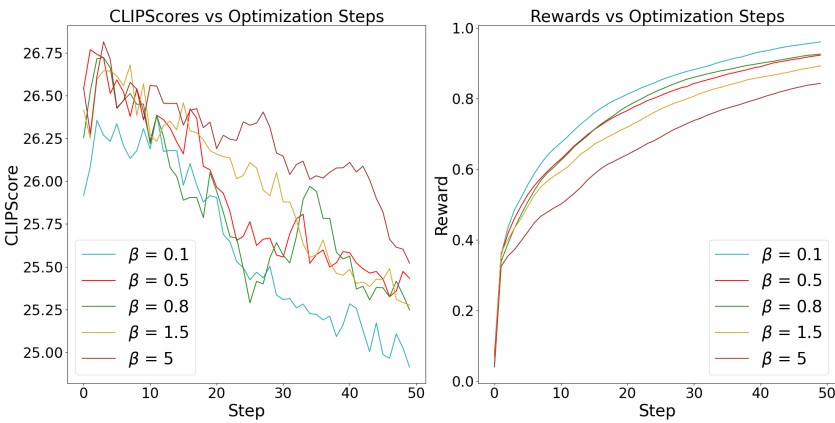

Figure 6: **Effect of different $\beta$ on CLIPScores (left) and rewards (right).** We obtain CLIPScores and rewards for all entries in the animals list in Appendix A.7. Curves depict the average result of five different seeds. The curves depict that lower $\beta$ exhibits higher rewards but lower CLIPScore. Higher $\beta$ tends to inadequately optimize for the reward.

### A.8 GPT-4O EVALUATION

We compute the win rate of generated images from our method and baselines, employing GPT-4o as the judge. The instruction prompt is as follows, where "PROMPT" and "TARGET REWARD" are placeholders.

---

**System Prompt**

You are a helpful, impartial, and precise assistant tasked with evaluating two images based on a given text prompt, a reward criterion, and specific quality guidelines. The text prompt is: generate an image of a "PROMPT". The reward criterion is: "TARGET REWARD." Please compare the two images and evaluate them based on the following:

1. Adherence to the text prompt: Assess which image better satisfies the intent of the text prompt by being meaningful and directly relevant to the described subject. Do not use the reward criterion to determine adherence to the prompt.

2. Satisfaction of the reward criterion: Determine which image more effectively meets the reward criterion.

Choose the image that maximizes the reward without compromising adherence to the prompt. With a lower priority, also ensure that:

1. The image does not have overly vibrant colors or extreme sharpening.
2. The lighting is balanced and natural.
3. The lighting is softer and more diffused.
4. The image appears realistic and natural, avoiding being overly saturated, sharpened, or monochromatic.
5. Shapes and objects are not overly simplified, distorted, or stylized.
6. The image is not completely white or black, unless explicitly required.
7. Relevant textures are present, and the image is not overly simplistic or missing essential details.
8. The composition is dynamic, creating a sense of energy and movement.

In the end, only output 1 if the first image is selected, or else output 2, and nothing else.

---

### A.9 PROOF OF PROPOSITION 1

*Proof.* To make our point that the change in the image distribution is unbounded with respect to the change in initial noise, we consider a simple setting with $z := x_T, \epsilon_T, \dots, \epsilon_1 \sim \mathcal{N}(0, \sigma^2 \mathbf{I})$, and the following random process

$$x_{t-1} = \theta x_t + \epsilon_t,$$

which holds from $t = T$ to 1. Now, applying the recursive relation starting from $x_T = z$, we can write

$$x_{T-1} = \theta z + \epsilon_T$$
$$x_{T-2} = \theta x_{T-1} + \epsilon_{T-1} = \theta^2 x_T + \theta \epsilon_T + \epsilon_{T-1}$$

$$\vdots$$

$$x_0 = \theta^T z + \sum_{k=1}^{T} \theta^{k-1} \epsilon_{T-k+1}. \tag{9}$$

This implies that the mean of $x_0$ is

$$\mathbb{E}[x_0|z] = \theta^T z, \tag{10}$$

and variance of $x_0$ can be written as

$$\text{Var}(x_0|z) = \text{Var}\left(\sum_{k=1}^{T} \theta^{k-1} \epsilon_{T-k+1}\right) = \sum_{k=1}^{T} \theta^{2k-2} \sigma^2$$

$$= \sigma^2 \frac{\theta^{2T} - 1}{\theta^2 - 1}, \tag{11}$$

and for simplicity we assume $|\theta| > 1$. From eqs. (10) and (11), we note that $x_0 \sim \mathcal{N}\left(\theta^T z, \sigma^2 \frac{\theta^{2T}-1}{\theta^2-1}\right)$. Suppose that we have two noises $z_1, z_2 \sim \mathcal{N}(0, \mathbf{I})$. Hence, for two close noise $z_1$ and $z_2$, we can write

$$d_{\text{KL}}[p(\cdot|z_1)\|p(\cdot|z_2)] = \frac{\theta^{2T}(\theta^2 - 1)}{2\sigma^2(\theta^{2T} - 1)} \cdot (z_1 - z_2)^2, \tag{12}$$

which holds from the closed-form expression of KL divergence between two Gaussian distributions. From the above expression in eq. (12), we note that the rate of change is controlled by the term $\frac{\theta^{2T}(\theta^2-1)}{2\sigma^2(\theta^{2T}-1)}$, which is controlled by the norm of neural network parameter $\theta$ which can be arbitrarily large in practice. For example, $\theta^2 \approx 1380000$ for SDv1.5.

$\square$

### A.10 DERIVATION OF THE PRACTICAL OBJECTIVE FOR DIFFERENTIABLE REWARDS

Here, we derive the score function approximation of our Lagrangian objective,

$$\mathcal{J}_{\text{MIRA}}(z, c) = \mathbb{E}_{x_0 \sim p_\theta(\cdot|z,c)}[r(x_0, c)] - \beta d_{\text{KL}}[p_\theta(x_0|z, c) \| p_\theta(x_0|z_0, c)]. \tag{13}$$

We consider the $d_{\text{KL}}$ term in eq. (13)

$$\mathcal{W} = d_{\text{KL}}[p_\theta(x_0|z, c) \| p_\theta(x_0|z_0, c)]. \tag{14}$$

We need to show that we can minimize $\mathcal{W}$ by minimizing

$$\mathcal{W}' \approx \mathbb{E}_{\tau \sim p_\theta(\cdot|z,c)}\left[\sum_{t=0}^{T-1} \sigma_t^2 \left(\|s(x_t|z_0, c)\|^2 - \|s(x_t|z, c)\|^2\right)\right], \tag{15}$$

with expectation over reverse trajectories $\tau$.

*Proof.* We start with eq. (14) and write it as follows:

$$\mathcal{W} = \mathbb{E}_{x_0 \sim p_\theta(\cdot|z,c)} \left[ \log \frac{p_\theta(x_0|z,c)}{p_\theta(x_0|z_0,c)} \right]. \tag{16}$$

Note that $d_{\mathrm{KL}}\big[p_\theta(x_0|z,c)\|p_\theta(x_0|z_0,c)\big] \leq d_{\mathrm{KL}}\big[p_\theta(\tau|z,c)\|p_\theta(\tau|z_0,c)\big]$, establishing an upper bound. Although this trajectory-based bound is formulated in the context of stochastic processes (SDEs), we note that the Probability Flow ODE (Song et al., 2021b) induces identical marginal distributions $p_t(x)$ at every timestep. Consequently, our score-based objective at the end of the derivation (eq. (26)) relies only on these marginals and remains valid for deterministic samplers (e.g., SDXL-Turbo, DDIM with $\eta = 0$) which share the same score function. Therefore,

$$\mathcal{W} \leq \mathcal{W}' = \mathbb{E}_{\tau \sim p_\theta(\cdot|z,c)} \big[ \log p_\theta(\tau|z,c) - \log p_\theta(\tau|z_0,c) \big]. \tag{17}$$

By linearity of expectation, this is

$$\mathcal{W}' = \mathbb{E}_{\tau \sim p_\theta(\cdot|z,c)} \big[ \log p_\theta(\tau|z,c) \big] - \mathbb{E}_{\tau \sim p_\theta(\cdot|z,c)} \big[ \log p_\theta(\tau|z_0,c) \big]. \tag{18}$$

Let us focus on the term $\mathbb{E}_{\tau \sim p_\theta(\cdot|z,c)}[\log p_\theta(\tau|z,c)]$, which we can expand as

$$\mathbb{E}_{\tau \sim p_\theta(\cdot|z,c)}[\log p_\theta(\tau|z,c)] = \mathbb{E}_{\tau \sim p_\theta(\cdot|z,c)} \left[ \log \prod_{t=0}^{T-1} p_\theta(x_t|x_{t+1},z,c) \right]. \tag{19}$$

We can expand eq. (19) as

$$\begin{aligned}
\mathbb{E}_{\tau \sim p_\theta(\cdot|z,c)}&[\log p_\theta(\tau|z,c)] \\
&= \mathbb{E}_{\tau \sim p_\theta(\cdot|z,c)} \left[ \log \left( \exp \left( -\sum_{t=0}^{T-1} \frac{\|x_t - \mu_\theta(x_{t+1},z,c)\|^2}{2\sigma_t^2} \right) \prod_{t=0}^{T-1} \frac{1}{\sigma_t\sqrt{2\pi}} \right) \right] \\
&= \mathbb{E}_{\tau \sim p_\theta(\cdot|z,c)} \left[ -\sum_{t=0}^{T-1} \frac{\|x_t - \mu_\theta(x_{t+1},z,c)\|^2}{2\sigma_t^2} \right] + \sum_{t=0}^{T-1} \log \frac{1}{\sigma_t\sqrt{2\pi}},
\end{aligned} \tag{20}$$

where $\mu_\theta(\cdot)$ is the predicted mean of the Gaussian at each step of the reverse process. We also know that for a Gaussian distribution,

$$\|s(x_t|z,c)\| = \|\nabla_{x_t} \log p(x_t|z,c)\| \approx \|\nabla_{x_t} \log p_\theta(x_t|x_{t+1};z,c)\|, \tag{21}$$

$$\|s(x_t|z,c)\| \approx \left\| \nabla_{x_t} \log \left( \frac{1}{\sigma_t\sqrt{2\pi}} \exp \left( -\frac{\|x_t - \mu_\theta(x_{t+1},t+1,z,c)\|^2}{2\sigma_t^2} \right) \right) \right\|, \tag{22}$$

$$\|s(x_t|z,c)\| \approx \left\| -\frac{x_t - \mu_\theta(x_{t+1},t+1,z,c)}{\sigma_t^2} \right\|. \tag{23}$$

Replacing eq. (23) in eq. (20), we get

$$\mathbb{E}_{\tau \sim p_\theta(\cdot|z,c)}[\log p_\theta(\tau|z,c)] \approx \mathbb{E}_{\tau \sim p_\theta(\cdot|z,c)} \left[ -\sum_{t=0}^{T-1} \|s(x_t|z,c)\|^2 \frac{\sigma_t^2}{2} \right] + \sum_{t=0}^{T-1} \log \frac{1}{\sigma_t\sqrt{2\pi}}. \tag{24}$$

Applying eq. (24) to eq. (17), we get

$$\mathcal{W}' \approx \frac{1}{2} \mathbb{E}_{\tau \sim p_\theta(\cdot|z,c)} \left[ \sum_{t=0}^{T-1} \left( \|s(x_t|z_0,c)\|^2 - \|s(x_t|z,c)\|^2 \right) \sigma_t^2 \right]. \tag{25}$$

We can use eq. (25) in eq. (13). Our final objective is

$$\mathcal{J}_{\text{MIRA}}(z, c) = \mathbb{E}_{x_0 \sim p_\theta(\cdot|z,c)}\big[r(x_0, c)\big] - \beta\mathbb{E}_{\tau \sim p_\theta(\cdot|z,c)}\bigg[\sum_{t=0}^{T-1}\sigma_t^2\left(\|s(x_t|z_0, c)\|^2 - \|s(x_t|z, c)\|^2\right)\bigg].$$

(26)

$\square$

### A.11 DERIVATION OF DPO OBJECTIVE

We begin with our original objective,

$$\max_z \mathcal{J}_{\text{MIRA}}(z, c) = \max_z \left[\mathbb{E}_{x_0 \sim p_\theta(\cdot|z,c)}\big[r(x_0, c)\big] - \beta d_{\text{KL}}\left[p_\theta(x_0|z, c)\,\|\,p_\theta(x_0|z_0, c)\right]\right].$$

(27)

Let $R(x_{0:T}, c)$ be the reward along the whole sampling trajectory such that

$$r(x_0, c) := \mathbb{E}_{x_{1:T} \sim p_\theta(\cdot|x_0,z,c)}\left[R(x_{0:T}, c)\right].$$

(28)

This means

$$\max_z \mathcal{J}_{\text{MIRA}}(z, c) = \max_z \left[\mathbb{E}_{\substack{x_0 \sim p_\theta(\cdot|z,c) \\ x_{1:T} \sim p_\theta(\cdot|x_0,z,c)}}\left[R(x_{0:T}, c)\right] - \beta d_{\text{KL}}\left[p_\theta(x_0|z, c)\,\|\,p_\theta(x_0|z_0, c)\right]\right]$$

$$= \min_z \left[-\mathbb{E}_{\substack{x_0 \sim p_\theta(\cdot|z,c) \\ x_{1:T} \sim p_\theta(\cdot|x_0,z,c)}}\left[R(x_{0:T}, c)\right] + \beta d_{\text{KL}}\left[p_\theta(x_0|z, c)\,\|\,p_\theta(x_0|z_0, c)\right]\right]$$

$$= \min_z \left[-\mathbb{E}_{x_0 \sim p_\theta(\cdot|z,c)}\left[\mathbb{E}_{x_{1:T} \sim p_\theta(\cdot|z,c)}\left[R(x_{0:T}, c)\right] - \beta\log\frac{p_\theta(\cdot|z, c)}{p_\theta(\cdot|z_0, c)}\right]\right]$$

$$= \min_z \left[\mathbb{E}_{x_0 \sim p_\theta(\cdot|z,c)}\left[\log\frac{p_\theta(\cdot|z, c)}{p_\theta(\cdot|z_0, c)} - \frac{1}{\beta}\mathbb{E}_{x_{1:T} \sim p_\theta(\cdot|x_0,z,c)}\left[R(x_{0:T}, c)\right]\right]\right]$$

$$\leq \min_z \left[\mathbb{E}_{x_{0:T} \sim p_\theta(\cdot|z,c)}\left[\log\frac{p_\theta(\cdot|z, c)}{p_\theta(\cdot|z_0, c)} - \frac{1}{\beta}R(x_{0:T}, c)\right]\right].$$

(29)

$$\max_z \mathcal{J}_{\text{MIRA}}(z, c) \leq \min_z \left[\mathbb{E}_{x_{0:T} \sim p_\theta(\cdot|z,c)}\log\frac{p_\theta(\cdot|z, c)}{\frac{1}{Z(c)}p_\theta(\cdot|z_0, c)\exp(\frac{1}{\beta}R(x_{0:T}, c))} - \log Z(c)\right],$$

(30)

$$\text{where} \quad Z(c) = \sum_{x_0} p_\theta(\cdot|z_0, c)\exp(\frac{1}{\beta}R(x_{0:T}, c)).$$

(31)

$$\max_z \mathcal{J}_{\text{MIRA}}(z, c) \leq \min_z \left[\mathbb{E}_{x_{0:T} \sim p_\theta(\cdot|z,c)}\log\frac{p_\theta(\cdot|z, c)}{p_\theta(\cdot|z^*, c)} - \log Z(c)\right],$$

(32)

$$\text{where} \quad p_\theta(\cdot|z^*, c) = \frac{1}{Z(c)}p_\theta(\cdot|z_0, c)\exp(\frac{1}{\beta}R(x_{0:T}, c)).$$

(33)

$$\max_{\mathbf{z}} \mathcal{J}_{\text{MIRA}}(z, c) \leq \min_z \left[d_{\text{KL}}[p_\theta(x_{0:T}|z, c)\,\|\,p_\theta(x_{0:T}|z^*, c)] - \log Z(c)\right].$$

(34)

Since $Z(c)$ does not depend on $z$, the minimum is achieved when the KL term is zero, which holds when the two distributions are identical, i.e.,

$$p_\theta(x_{0:T}|z^*, c) = \frac{1}{Z(c)}p_\theta(x_{0:T}|z_0, c)\exp(\frac{1}{\beta}R(x_{0:T}, c)).$$

(35)

Taking the logarithm on both sides and rearranging the terms, we get

$$R(x_{0:T}, c) = \beta\log Z(c) + \beta\log\frac{p_\theta(x_{0:T}|z, c)}{p_\theta(x_{0:T}|z_0, c)}.$$

(36)

This means, from eq. (28),

$$r(x_0, c) = \beta \log Z(c) + \beta \mathbb{E}_{x_{1:T} \sim p_\theta(\cdot|x_0, z, c)} \left[ \log \frac{p_\theta(x_{0:T}|z, c)}{p_\theta(x_{0:T}|z_0, c)} \right]. \tag{37}$$

Assume that we have access to a dataset $\{(z^{(1)}, x_0^{(1)}), (z^{(2)}, x_0^{(2)}), c\}_{i=1}^N$, where the generated image $x_0^{(1)}$ is preferred over $x_0^{(2)}$. Under the Bradley-Terry model, the human preference distribution can be written as:

$$P^*(x_0^{(1)} \succ x_0^{(2)}|z^{(1)}, z^{(2)}, c) = \frac{\exp(r^*(x_0^{(1)}, c))}{\exp(r^*(x_0^{(1)}, c)) + \exp(r^*(x_0^{(2)}, c))}$$

$$= \sigma(r^*(x_0^{(1)}, c) - r^*(x_0^{(2)}, c)). \tag{38}$$

Framing the problem as binary classification, we have the negative log-likelihood loss as:

$$\mathcal{L}_{\text{MIRA-DPO}}(z^{(1)}, z^{(2)}, c) = -\mathbb{E}_{\substack{x_0^{(1)} \sim p_\theta(\cdot|z^{(1)}, c) \\ x_0^{(2)} \sim p_\theta(\cdot|z^{(2)}, c)}} [\log \sigma(r^*(x_0^{(1)}, c) - r^*(x_0^{(2)}, c))]. \tag{39}$$

Using eq. (37) in eq. (39), we get

$$\mathcal{L}_{\text{MIRA-DPO}}(z^{(1)}, z^{(2)}, c) =$$

$$-\mathbb{E}_{\substack{x_0^{(1)} \sim p_\theta(\cdot|z^{(1)}, c) \\ x_0^{(2)} \sim p_\theta(\cdot|z^{(2)}, c)}} \log \sigma \left( \beta \mathbb{E}_{\substack{x_{1:T}^{(1)} \sim p_\theta(\cdot|x_0^{(1)}, z^{(1)}, c) \\ x_{1:T}^{(2)} \sim p_\theta(\cdot|x_0^{(2)}, z^{(2)}, c)}} \left[ \log \frac{p_\theta(x_{0:T}^{(1)}|z^{(1)}, c)}{p_\theta(x_{0:T}^{(1)}|z_0^{(1)}, c)} - \log \frac{p_\theta(x_{0:T}^{(2)}|z^{(2)}, c)}{p_\theta(x_{0:T}^{(2)}|z_0^{(2)}, c)} \right] \right). \tag{40}$$

Since we assume $x_0^{(1)} \succ x_0^{(2)}$, we know $x_0^w := x_0^{(1)}$ and $x_0^l := x_0^{(2)}$. We can rewrite eq. (40) as

$$\mathcal{L}_{\text{MIRA-DPO}}(z^w, z^l, c) =$$

$$-\mathbb{E}_{\substack{x_0^w \sim p_\theta(\cdot|z^w, c) \\ x_0^l \sim p_\theta(\cdot|z^l, c)}} \log \sigma \left( \beta \mathbb{E}_{\substack{x_{1:T}^w \sim p_\theta(\cdot|x_0^w, z^w, c) \\ x_{1:T}^l \sim p_\theta(\cdot|x_0^l, z^l, c)}} \left[ \log \frac{p_\theta(x_{0:T}^w|z^w, c)}{p_\theta(x_{0:T}^w|z_0^w, c)} - \log \frac{p_\theta(x_{0:T}^l|z^l, c)}{p_\theta(x_{0:T}^l|z_0^l, c)} \right] \right), \tag{41}$$

which can be conceptually understood as the standard DPO loss, $\mathcal{L} = -\mathbb{E}[\log \sigma(\hat{r}^w - \hat{r}^l)]$. Here, we define the implicit, regularized MIRA reward $\hat{r}$ as the log-ratio of the noise-optimized and reference model probabilities, which is practically realized using our score-based surrogate. Hence, this is our formulation in the main paper.

**Practically, we can use our score function surrogate.**

By Jensen's inequality,

$$\mathcal{L}_{\text{MIRA-DPO}}(z^w, z^l, c) \le -\log \sigma \left( \beta \mathbb{E}_{\substack{x_{0:T}^w \sim p_\theta(\cdot|z^w, c) \\ x_{0:T}^l \sim p_\theta(\cdot|z^l, c)}} \left[ \log \frac{p_\theta(x_{0:T}^w|z^w, c)}{p_\theta(x_{0:T}^w|z_0^w, c)} - \log \frac{p_\theta(x_{0:T}^l|z^l, c)}{p_\theta(x_{0:T}^l|z_0^l, c)} \right] \right). \tag{42}$$

Therefore,

$$\mathcal{L}_{\text{MIRA-DPO}}(z^w, z^l, c) \le -\log \sigma \left( \beta d_{\text{KL}} \left[ p_\theta(x_{0:T}^w|z^w, c) \| p_\theta(x_{0:T}^w|z_0^w, c) \right] \right.$$

$$\left. - \beta d_{\text{KL}} \left[ p_\theta(x_{0:T}^l|z^l, c) \| p_\theta(x_{0:T}^l|z_0^l, c) \right] \right).$$

As we have shown previously in A.10, we can approximate the KL terms. Our final loss is

$$\mathcal{L}_{\text{MIRA-DPO}}(z^w, z^l, c)$$

$$= -\log \sigma \left( \beta \mathbb{E}_{x_{0:T}^w \sim p_\theta(\cdot|z^w, c)} \left[ \sum_{t=0}^{T-1} \sigma_t^2 \left( \|s(x_t^w|z_0^w, c)\|^2 - \|s(x_t^w|z^w, c)\|^2 \right) \right] \right. \tag{43}$$

$$\left. - \beta \mathbb{E}_{x_{0:T}^l \sim p_\theta(\cdot|z^l, c)} \left[ \sum_{t=0}^{T-1} \sigma_t^2 \left( \|s(x_t^l|z_0^l, c)\|^2 - \|s(x_t^l|z^l, c)\|^2 \right) \right] \right).$$

