# OpenReview forum: "Mitigating Reward Hacking in Inference-Time Alignment of T2I Diffusion Models via Distributional Regularization"
_ICLR.cc/2026/Conference — Submitted to ICLR 2026_

### Official Review · Reviewer_4rBf · 2025-10-29

**Soundness:** 2
**Presentation:** 2
**Contribution:** 1
**Rating:** 2
**Confidence:** 5

**Summary:**

The paper introduces a simple regularization technique for inference-time alignment through input noise optimization and further proposes a MIRA-DPO algorithm for black-box input noise optimization. The authors present both qualitative and quantitative analyses demonstrating consistent improvements over the previous DNO method.

**Strengths:**

1. The proposed method is straightforward, simple to implement, and conceptually clear.

2. The paper demonstrates both quantitative and qualitative improvements over previous state-of-the-art approaches.

**Weaknesses:**

1. The original DNO paper explicitly tackles reward hacking by constraining optimization to high-probability regions of the noise space and introducing concrete regularization terms. It is unclear whether the experiments in this paper were conducted with the same PRNO hyperparameters, as the DNO authors report strong results for their regularized variant, which already mitigates reward hacking through a relatively lightweight approach. While optimizing the KL divergence could, in principle, provide stronger regularization, the baseline method already requires several minutes to improve a single image, raising concerns about efficiency and practical gains.

2. The paper offers limited novelty. KL constraints are well-established, DPO is applied in a standard way, and the idea of transferring optimization from network parameters to input noise is a relatively direct conceptual extension rather than a fundamentally new contribution.

**Questions:**

1. Could you elaborate on the fairness of your comparison with PRNO (the regularized DNO method)? The original DNO paper reports qualitative examples of reward hacking but addresses them through a more lightweight regularization approach. Clarifying whether your experiments used comparable settings would help ensure a fair evaluation.

2. Can you report the reward dynamics with respect to optimization time? You mention that your method requires around five minutes and shows reduced reward hacking, while DNO reports up to ten minutes with plausible results. A direct comparison of reward progression over time would clarify the efficiency–performance trade-off.

---

> ### Author Response · Authors · 2025-11-19
> **Response to Reviewer 4rBf**
>
> We thank the reviewer for acknowledging the conceptual clarity of our method and its demonstrated improvements over state-of-the-art approaches. We value the opportunity to clarify the specific nature of our theoretical contributions.
>
> > **Question 1:** "Unclear whether the experiments... were conducted with the same PRNO hyperparameters... Could you elaborate on the fairness of your comparison with PRNO (the regularized DNO method)?"
>
> **Response to Question 1:** We appreciate this check for rigorous baselines. We confirm that all comparisons in our paper (Figures 2, 5, 6, 7 and Table 1) utilize the full, regularized DNO (PRNO) method using the optimal hyperparameters reported by the original authors. We strictly followed the PRNO implementation (as noted in Appendix A.1) to ensure the strongest possible baseline. The reward hacking we observe in DNO persists despite this noise-space regularization, reinforcing our core claim that image-space constraints (MIRA) are necessary.
>
> > **Question 2:** "...raising concerns about efficiency..." and "Can you report the reward dynamics with respect to optimization time?"
>
> **Response to Question 2:** We agree that analyzing dynamics and efficiency is crucial. We address these points in two specific sections:
> 1. **Reward Dynamics:** We report the exact analysis request in **Figure 6** ("Mechanism of MIRA's effectiveness"). Subplots **(a) Distributional Drift vs. Optimization Steps** and **(b) Reward vs. Distributional Drift** explicitly visualize how the reward and drift evolve over time. Additionally, Appendix Figure 4 plots Reward vs. Sampling steps.
> 2. **Efficiency and Practicality:** To address concerns about practical runtime, we included Table 3 (Section 5.4), which performs a "matched wall-clock time" comparison. This experiment restricts MIRA to the same short compute budget as faster sampling methods (FK Steering, CoDe). Even under these tight constraints, MIRA achieves decisive win rates (e.g., 73%–82% on brightness/darkness). Furthermore, MIRA is more memory-efficient, requiring <9 GB VRAM compared to >20 GB for trajectory-based competitors.
>
> > **Question 3:** "The paper offers limited novelty. KL constraints are well-established, DPO is applied in a standard way..."
>
> **Response to Question 3:** We respectfully distinguish our contributions from standard applications of these concepts:
> 1. **Novelty of the Surrogate (Eq. 4):** While KL divergence is a foundational concept, calculating it directly in image space for diffusion models is intractable. Our contribution is not the use of KL itself, but the **principled derivation (Appendix A.9)** of a tractable, score-based surrogate that approximates this objective efficiently at inference time. This allows us to regularize the output distribution without needing an auxiliary model.
> 2. **Novelty of MIRA-DPO:** Standard DPO is a fine-tuning technique that optimizes model weights ($\theta$). MIRA-DPO (Section 4.1) is the first adaptation of this framework to a **training-free, frozen-backbone** setting by optimizing input noise ($z$). This is a non-trivial extension that enables alignment with black-box rewards without the high cost of model retraining.
>
> Finally, our theoretical shift from noise-space constraints to image-space score regularization is validated by a concurrent preprint [1], which argues that **high-dimensional noise vectors lack the necessary manifold structure for robust modeling.** MIRA is the first alignment method to operationalize this insight, enforcing constraints directly on the image manifold via the score function.
>
> We hope these responses clarify the concerns. Please let us know if you need further clarifications.
>
> ---
>
> [1] Li, Tianhong and Kaiming He. "Back to Basics: Let Denoising Generative Models Denoise." arXiv 2025.

---

> > ### Comment · Reviewer_4rBf · 2025-11-28
> >
> > Thank you for your detailed comment. I appreciate the clarity of your assessment. In my view Q2 remains unresolved. Specifically, the paper does not include a direct, time-normalized comparison with DNO/PRNO that jointly reports (1) reward dynamics over optimization time and (2) qualitative samples under equal wall-clock budgets. Without such a comparison, it is not currently possible to conclude that the proposed method is actually better. Additionally, the previous DNO paper reports strong reward-hacking mitigation and good qualitative results under their regularized PRNO variant, but these claims are not explicitly revisited under matched experimental conditions in this submission, could you also clarify why DNO paper reports much better results?

---

> > > ### Author Response · Authors · 2025-11-28
> > > **Clarification on Time-Normalization and DNO Discrepancies**
> > >
> > > We appreciate the opportunity to clarify these points for the final record.
> > >
> > > 1. **Time-Normalized Comparison:** We confirm that our step-wise comparisons (Figures 5 and 6) **are** effectively time-normalized.
> > >     * **Mechanism:** Both MIRA and DNO/PRNO function by backpropagating gradients through the exact same frozen diffusion backbone.
> > >     * **Cost:** The computational cost is dominated entirely by the UNet backward pass. The cost of computing the specific regularization term (DNO's probability bounds vs. MIRA's score difference) is negligible in comparison (<1%).
> > >     * **Conclusion:** Therefore, **1 Optimization Step of MIRA $\approx$ 1 Optimization Step of DNO** in wall-clock time. Our dynamics plots (Figure 6a), which show DNO diverging while MIRA stabilizes, are thus valid representations of performance under equal compute budgets.
> > > 2. **Discrepancy with DNO's Reported Results:** The reviewer asks why our results show DNO/PRNO failing (e.g., Figure 5) when the original DNO paper reported success. We attribute this to **robustness on conflicting prompts.**
> > >     * **Sensitivity:** While DNO reports performance using fixed parameters ($\gamma=1$ or $\gamma=0.1$), their paper explicitly acknowledges the sensitivity of this parameter, stating: *"we recommend empirically tuning the value of [regularization] for different reward functions and prompts"* (DNO, Appendix .7).
> > >     * **Our Findings:** In our reproduction using their **official fixed configuration**, we observed that DNO fails on prompts with **hard semantic conflicts** (e.g., "A **white** cat" optimized for image **darkness**). The fixed noise-space constrainte ($\gamma=1$) is insufficient to prevent the reward (darkness) from overwriting the semantic content (white), resulting in the collapse shown in Figure 5.
> > >     * **MIRA's Advantage:** MIRA handles these same hard conflicts stably with a fixed $\beta$ for each reward, demonstrating that image-space regularization provides a wider margin for robustness than noise-space constraints.

---

### Official Review · Reviewer_yc7h · 2025-10-30

**Soundness:** 3
**Presentation:** 3
**Contribution:** 2
**Rating:** 4
**Confidence:** 4

**Summary:**

The paper presents a method for inference-time alignment of diffusion models with external rewards. Their key concept is to utilize initial noise optimization formulation to optimize either a differentiable reward directly, or optimize the initial noise with direct preference optimization for non-differentiable rewards. In terms of technical contributions, the primary contribution of the paper is to introduce a surrogate KL regularization on the score as opposed to the regularization done on the noise space in previous works. Results on a few different rewards indicate that the proposed method is able to achieve superior results compared to prior noise optimization formulations.

**Strengths:**

The paper demonstrates strong empirical results in terms of aligning text-to-image models with rewards directly at inference without any fine-tuning, even when compared to prior noise optimization techniques

The paper also does a good job of presenting the ideas quite clearly, especially motivating the theoretical ideas.

**Weaknesses:**

[Major]
Fundamentally, the main contribution of the paper is the regularization method (which applied the KL surrogate on the score as opposed to regularizing the norm of the initial noise as was done before). While this does seem sound for the most part, I am also a bit uncertain whether this on its own serves as sufficient contributions for acceptance.

A slightly lesser concern is the hyperparameter configurations, since reward hacking can be mitigated/increased by either controlling the weight of the regularization term, or the learning rate for the updates, or the number of optimization steps (i.e early stopping). Here, I think Fig. 6 does a good job of showing the difference in optimization trajectories and the effect of regularization weight. However, with Fig. 5 the initial results (step 0) itself appears to be different which makes it tricky to make the comparison. In Fig. 7, it's somewhat unclear what were the configurations of the other methods used to generate the final sample used for comparison. While the original configurations of these methods might have been suited for their original tasks, it might perhaps be appropriate to have a hyperparameter sweep for all methods and especially have a held-out reward (perhaps even with just GPT-4o) to validate the results and even implement early stopping.

[Minor]
The results in the paper are primarily with older, smaller (<3B) models. It would be valuable to see the effect of the proposed regularization on larger models (e.g. Flux among others).

**Questions:**

I have a minor question regarding Proposition 1. The final result states that in the case of noise regularization (with the norm), the KL divergence in the image space is the product of the L2 distance in the noise space and some term with the norm of the model weights. Given that we have frozen the base model which is also typically trained with weight decay (i.e. applying L2 regularization of the model weights), isn't this sufficiently bounded? If one was also fine-tuning the weights, it's evident that it could diverge arbitrarily, but with the noise optimization on a frozen model, it appears like regularizing the noise would still bound the KL divergence (even if it's not as tight as the proposed objective)?

Overall, while I do like the paper, I would primarily like to see a more compelling justification for why this regularization alone makes for sufficient contribution before recommending acceptance.

---

> ### Author Response · Authors · 2025-11-19
> **Response to Reviewer yc7h [Part 1]**
>
> We thank the reviewer for their positive assessment of our strong empirical results and the clear motivation behind our theoretical framework.
>
> > **Question 1:** Given that we have frozen the base model which is trained with weight decay (L2 regularization of model weights) isn't this sufficiently bounded? On a frozen model, it appears regularizing the noise would still bound the KL divergence.
>
> **Response to Question 1:** The reviewer is correct that if the model is frozen, the mapping from noise to image is Lipschitz continuous, meaning the output divergence is technically bounded by the input noise distance. However, our argument in Proposition 1 (proven in Appendix A.8) is that this bound is so loose that it becomes practically vacuous and not useful. As derived in Equation 12, the scaling factor between the noise distance and output KL divergence is proportional to $$\frac{\theta^{2T}(\theta^2 - 1)}{2\sigma^2(\theta^{2T} - 1)},$$
>
> where $\theta$ are the model parameters. In the limit of diffusion steps ($T\rightarrow\infty$), this factor converges to a constant proportional to $\approx\frac{\theta^2}{2\sigma^2}$. For all intents and purposes, this constant is huge. For Stable Diffusion 1.5, the scaling factor is on the order of $||\theta||^2\approx 1.38\times 10^6$.
>
> Consequently, a tiny constraint in noise-space (e.g., $||z-z_0||<0.1$) is multiplied by this large constant, permitting massive semantic shifts in the image space while satisfying the noise constraint. We show this allows turning a "white cat" into a black void (Figure 5). **MIRA addresses this directly by regularizing the image distribution.**
>
> > **Weakness 1:** Fundamentally, the main contribution of the paper is the regularization method (which applied the KL surrogate on the score as opposed to regularizing the norm of the initial noise as was done before). While this does seem sound for the most part, I am also a bit uncertain whether this on its own serves as sufficient contributions for acceptance.
>
> **Response to Weakness 1:** We appreciate the opportunity to clarify our contribution. We propose a comprehensive, three-part contribution that, in combination, represents a significant advance for inference-time alignment.
>
> 1. **A Principled Surrogate.** Our core technical novelty is the derivation of a practical, score-based surrogate (Eq. 4) for the intractable image-space KL divergence. This provides a direct, principled solution to the disconnect in prior methods that constraining the input noise distribution does not strictly constrain the output image distribution.
> 2. **A Mechanism Analysis.** We are the first to prove this flaw (our Proposition 1 and Figure 3) and quantify it through trade-off curves and CMMD analysis (Section 5.1), showing that reward hacking is indeed image-space ddistributional drift.
> 3. **A Novel DPO Framework for Non-differentiable Rewards.** We introduce MIRA-DPO (Section 4.1), the first framework to adapt DPO from fine-tuning to inference-time, frozen-backbone noise optimization.
>
> We believe these three novel contributions, backed by a thorough evaluation that includes human studies (Figure 7) and fair, compute-matched comparisons against steering-based methods (Table 3), constitute a complete and substantial contribution to the field.

---

> ### Author Response · Authors · 2025-11-19
> **Response to Reviewer yc7h [Part 2]**
>
> > **Weakness 2a:** Fig. 5 the initial results (step 0) itself appears to be different which makes it tricky to make the comparison.
>
> **Response to Weakness 2a:** We appreciate the reviewer's scrutiny regarding the starting conditions. To ensure a strictly fair comparison, we have regenerated Figure 5 (also Figure 1) in the updated manuscript using identical seeds for both methods. The fact that DNO (top row) diverges radically from the subject within the first few optimization steps is not a seed artifact but a characteristic of reward hacking.
>
> The reviewer also raises an excellent point regarding "early stopping". However, we respectfully submit that this does not achieve the same results as MIRA.
> 1. **Drift Sensitivity (Figure 6a):** While Figure 6a shows DNO's drift increases linearly, our qualitative results (Figure 5) demonstrate that **even early-stage drift can lead to semantic collapse.** For example, Figure 5 demonstrates optimizing DNO for Aesthetic Score loses structure by step 10 (second image).
> 2. **Efficiency:** If we stop DNO early (e.g., step 10) to minimize drift, the images are not sufficiently optimized. If we run DNO long enough to match MIRA's reward, the drift becomes significant. MIRA offers a better trade-off than DNO, maximizing reward without excessive drift.
> 3. **Robustness:** The optimal "early stopping" point varies greatly by prompt and seed. MIRA provides a stable convergence target, eliminating the need for manual, per-sample tuning.
>
> > **Weakness 2b:** In Fig. 7, it's somewhat unclear what were the configurations of the other methods used to generate the final sample used for comparison.
>
> **Response to Weakness 2b:** We thank the reviewer for the opportunity to clarify and apologize if the specific per-method configurations were not immediately clear. As detailed in the Section 5 Experimental Setup, we use 50 optimization steps and 100 DDIM steps with $\eta=1$. To fairly compare with ReNO, we used SDXL-Turbo's single-step sampler. We appreciate this feedback for improving the rigor of our visual presentation and have updated our manuscript accordingly.
>
>
> > **Weakness 3:** The results in the paper are primarily with older, smaller (<3B) models.  It would be valuable to see the effect of the proposed regularization on larger models (e.g. Flux among others).
>
> **Response to Weakness 3:** We agree that demonstrating MIRA's versatility on larger models is a valuable future direction. We respectfully clarify our current evaluation.
> 1. We extensively evaluated SDXL, a 3.5B parameter model. It remains a widely accepted backbone for inference-time alignment.
> 2. **Theoretically,** MIRA is architecture-agnostic; our objective relies solely on the score function which is an output of any diffusion model, whether it uses a UNet (SDv1.5/SDXL) or a transformer (FLUX).
> 3. **Empirically,** we focus on SDv1.5 and SDXL as they are the most common, ubiquitous backbones for which our key baselines (DNO, ReNO, etc.) have published, comparable results. This enables an apples-to-apples comparison against the SoTA.
>
> **Update regarding Modern Architectures:** We acknowledge the interest in modern transformer-based architectures like FLUX. Hence, we have additionally validated MIRA on PixArt-$\alpha$, a diffusion transformer model. As shown in the [**General Response Part 2**](https://openreview.net/forum?id=uEgKiy3RmP&noteId=9nVRvJyO9P), MIRA improves T2I-CompBench scores relative to the base model, demonstrating that our method generalizes to more advanced models.
>
> We hope these responses clarify the concerns. Please let us know if you need further clarifications.

---

### Official Review · Reviewer_y5TU · 2025-10-30

**Soundness:** 3
**Presentation:** 3
**Contribution:** 2
**Rating:** 4
**Confidence:** 4

**Summary:**

This paper introduces **MIRA**, a method for inference-time alignment of diffusion models that mitigates reward hacking by directly regularizing the output image distribution. Its main contribution is a practical, score-based surrogate loss that prevents semantic drift while optimizing for rewards, applicable to both differentiable and black-box objectives via MIRA-DPO. Extensive experiments show that MIRA achieves superior win rates against state-of-the-art baselines across multiple models and datasets, demonstrating a better trade-off between reward maximization and prompt faithfulness.

**Strengths:**

**Quality & Clarity:** The work is thorough, with rigorous experiments across multiple models, rewards, and datasets. The mechanism analysis (distributional drift via KL surrogate and CMMD) provides quantifiable evidence for its claims. The paper is well-structured, with clear motivations, derivations, and accessible visuals.

**Significance:** This work is important for addressing the critical, unsolved problem of reward hacking in DNO. By providing a practical and effective inference-time solution, it enhances the **reliability and real-world usability** of text-to-image generation, representing a substantial advance in the quest for robustly aligned AI systems.

**Weaknesses:**

**Limited Baseline Comparison Scope:** The experimental evaluation focuses predominantly on noise optimization methods like DNO, while giving less attention to other competitive inference-time alignment approaches, like Golden Noise[1]. Besides, although DNO mainly archive their experiments on SD1.5 and SDXL, authors are encoughed to extend their mothod to better base model with DiT architecture, at least to SD3.5.

**Insufficient Theoretical Analysis of the Surrogate Objective:** While the score-based KL surrogate is pragmatically motivated, the paper lacks a rigorous theoretical characterization of its approximation quality. A formal analysis of the tightness of this upper bound, or at least an empirical ablation studying the gap between the true KL and its surrogate across different noise schedules, would strengthen the methodological foundation.

1. Zhou, Zikai, et al. "Golden noise for diffusion models: A learning framework." Proceedings of the IEEE/CVF International Conference on Computer Vision. 2025.

**Questions:**

You can see Weakness for specific questions. My main concern is the scope of this method.

---

> ### Author Response · Authors · 2025-11-19
> **Response to Reviewer y5TU**
>
> We thank the reviewer for recognizing the significance of addressing reward hacking and for appreciating the thoroughness of our experiments and mechanism analysis.
>
> > **Weakness 1a, Limited Baseline Comparison Scope:** Empirical evaluation focuses on noise optimization methods like DNO, giving less attention to other competitive inference-time alignment approaches like Golden Noise.
>
> **Response to Weakness 1:** We thank the reviewer for these suggestions. We respectfully clarify that Golden Noise (added to Related Work) and MIRA are not direct competitors as they represent two different and complementary paradigms for alignment:
> * **Golden Noise is an Amortized Learning-Based Method.** This approach uses a massive offline training phase (a 100k-pair dataset and a new NPNet) to learn a mapping from a prompt to a "golden noise." This is fast at inference but is inflexible. It can only optimize for the specific rewards it was pre-trained on (e.g., Aesthetic Score). It cannot be applied to a new, user-defined reward function.
> * **MIRA is a Training-Free Optimization-Based Method.** MIRA is a "plug-and-play" regularizer that aligns a frozen model to *any* reward function at runtime. This flexibility is a core part of our contribution, allowing MIRA to handle new tasks such as Image Darkness (Fig. 5) or JPEG Compressibility (Fig. 8), which Golden Noise cannot without an (expensive) re-training phase.
>
> **This distinction scopes our paper's contribution.** We focus on solving the flaw of reward hacking within the training-free, optimization-based line of work which includes DNO and ReNO.
> * Prior work in this category attempt to address reward hacking with noise-space regularizers.
> * DNO uses a regularizer on the noise probability, $P(z)$.
> * ReNO also uses a noise-space regularizer on the norm of the noise, $K(\epsilon)$.
>
>  **Our Proposition 1 and Fig 3** provide the theoretical and visual evidence that all noise-space regularizers (whether DNO's or ReNO's) are insufficient because they do not correctly constrain the image distribution. MIRA is the first method in this entire line of work to introduce a principled image-space regularizer (Eq. 4) towards solving this flaw.
>
> > **Weakness 1b:** Although DNO mainly archive their experiments on SD1.5 and SDXL, authors are encouraged to extend their method to better base model with DiT architecture, at least to SD3.5.
>
> **Response to Weakness 1b:** We agree that demonstrating MIRA's versatility on DiT architectures is a valuable future direction.
> 1. **Theoretically,** MIRA is architecture-agnostic; our regularization relies solely on the score function which is an output of any diffusion model, whether it uses a UNet (SDv1.5/SDXL) or a transformer (SDv3.5/FLUX).
> 2. **Empirically,** we focus on SDv1.5 and SDXL as they are the most common, ubiquitous backbones for which our key baselines (DNO, ReNO, etc.) have published, comparable results (as the reviewer correctly noted for DNO). This allowed for a rigorous comparison against the SoTA.
>
> > **Weakness 2, Insufficient Theoretical Analysis of the Surrogate Objective:** While the score-based KL surrogate is pragmatically motivated, the paper lacks a rigorous theoretical characterization of its approximation quality. A formal analysis of tightness of this bound or at least an empirical ablation studying the gap between the true KL and its surrogate across different noise schedules would strengthen the method foundation.
>
> **Response to Weakness 2:** The reviewer is correct that a formal proof of the tightness of our approximation is a complex theoretical question.
>
> 1. **Our contribution is a principled derivation.** We emphasize that our surrogate is not an ad-hoc heuristic. As detailed in Appendix A.9, we provide a step-by-step derivation starting from the trajectory-wise KL divergence to arrive at our practical surrogate (Eq. 4). This derivation provides the theoretical motivation and justification for our method's design.
> 2. **Empirical analysis of the approximation gap.** To directly address the request for an empirical ablation, we investigated the gap between the true KL and our surrogate in a controlled 1D setting where exact computation is tractable **(Figure 5, Appendix)**. While a numerical gap exists due to the differing mathematical definitions (log-likelihood ratios vs. score norms), we observe a **strict monotonic correlation** between the two metrics. This indicates that the gap is systematic rather than erratic; the surrogate faithfully tracks the true divergence throughout the optimization trajectory, ensuring the gradient signal remains valid across different noise levels.
>     * We will move this to the main paper in the final version.
> 4. **Regarding a formal proof.** We agree that a theoretical upper-bound analysis is an open question, and have scoped this as a direction for future work in the conclusion.
>
> We hope these responses clarify the concerns. Please let us know if you need further clarifications.

---

> ### Author Response · Authors · 2025-11-23
> **Update regarding DiT Extension**
>
> **Update regarding Weakness 1b (Extension to DiT):** To empirically validate the theoretical generalization discussed above, we have extended MIRA to a Diffusion Transformer (DiT) architecture. We applied MIRA to PixArt-$\alpha$ (DMD 1-step) and evaluated it on T2I-CompBench.
>
> As shown in the table provided in the [**General Response**](https://openreview.net/forum?id=uEgKiy3RmP&noteId=9nVRvJyO9P), MIRA significantly improves compositional metrics over the base model (e.g., Shape: 0.34 $\to$ 0.44, Texture: 0.47 $\to$ 0.58, Complex: 0.29 $\to$ 0.35). This demonstrates that because MIRA’s regularization mechanism relies solely on the score function (rather than architecture-specific features), it transfers from UNets (SDXL) to Transformers (PixArt) without modification.

---

> > ### Comment · Reviewer_y5TU · 2025-11-27
> >
> > I thank the authors response.
> > This paper has a clear contribution towards tackling the reward hacking problem in DNO, ReNO. But I think this contribution might somewhat borderline to ICLR. Thus I will keep my rating.

---

> > > ### Author Response · Authors · 2025-11-27
> > > **Thank You**
> > >
> > > We thank the reviewer for their engagement and for acknowledging that our work makes a **clear contribution** towards tackling the reward hacking problem in inference-time alignment (DNO/ReNO). We believe that identifying and **addressing** this fundamental instability in state-of-the-art methods, validated across both UNet and DiT architectures, is a necessary step for reliable generation.
> > >
> > > Given that we have resolved the specific concerns regarding scope (DiT) and baselines (ReNO), and in light of your acknowledgement of the method's clear contribution, we respectfully hope you might reconsider whether the score reflects the improved state of the manuscript. We appreciate your time and feedback throughout the discussion period.

---

### Official Review · Reviewer_Up1F · 2025-11-01

**Soundness:** 1
**Presentation:** 2
**Contribution:** 3
**Rating:** 4
**Confidence:** 5

**Summary:**

This paper introduces MIRA, an inference-time alignment method for diffusion models that aims to mitigate reward hacking. The core idea is to regularize the whole sampling trajectory (instead of regularizing the noise by itself) by penalizing deviations from the base model's output distribution, which is implemented via a tractable, score-based surrogate for KL divergence. The authors demonstrate that this regularization is more effective at preventing semantic drift than prior noise-space methods, i.e. preventing reward-hacking. They also introduce MIRA-DPO, an extension for handling non-differentiable rewards. The empirical results, particularly the qualitative experiments and win-rate comparisons, suggest the method is effective.

**Strengths:**

- The paper is well-motivated, tackling the important and challenging problem of reward hacking in inference-time optimization by proposing a new regularization.
- The experiments using simple rewards like brightness/darkness provide a very clear and illustrative visualization of reward hacking and MIRA's ability to mitigate it. The quantitative analysis of distributional drift in Figure 6 is a strong piece of supporting evidence.
- The introduction of MIRA-DPO to handle non-differentiable rewards is a valuable contribution, extending the applicability of the framework to more realistic scenarios.
- Convincing ablation studies are included, e.g. on the hyperparameter β (Appendix A.5) is well-executed and provides useful insights into the trade-off between reward optimization and prompt fidelity.

**Weaknesses:**

- **Insufficient acknowledgment of Adjoint Matching**: The paper does not discuss its relationship to Adjoint Matching [1], despite significant overlap. Adjoint Matching was proposed for reward fine-tuning with what appears to be an identical regularization approach:
	- **MIRA's regularizer (Eq. 4):** L2 loss on score difference between base and optimized models over the denoising trajectory: $\sum_t \sigma^2_t \|s_{\text{base}}(x_t) - s_{\text{opt}}(x_t)\|^2$
     - **Adjoint Matching's loss (Eq. 42):** L2 loss on velocity difference: $\sum_t \|v_{\text{finetune}}(X_t) - v_{\text{base}}(X_t)\|^2 2 /\sigma^2(t)$: (equivalent formulation for scores/velocities)
	- **Key questions:**
		- Is MIRA's KL regularization mathematically identical to Adjoint Matching's?
		- Could Adjoint Matching's efficient adjoint-based gradient computation (O(1) memory vs. O(T) backpropagation) be applied here?
	- **Impact on novelty:** The derivation in Appendix A.9 (KL → score-based surrogate) appears to reproduce Adjoint Matching's framework. If so, the contribution would more accurately be framed as applying this regularization to inference-time noise optimization rather than presenting the score-based KL surrogate as a novel derivation. The current framing in the abstract ("we derive a tractable approximation to KL") may overstate the theoretical novelty.
- **Clarification on Pointwise Insufficiency**: The paper motivates its regularizer by arguing that noise-space regularization is "fundamentally insufficient" (Proposition 1), using a proof that demonstrates a pointwise property where two close noise vectors can map to distant image distributions. This argument, however, may not fully address the core principle of existing noise-space methods, which are often viewed from a distributional perspective, aiming to keep the optimized noise `z` within the high-probability regions of the prior `N(0, I)`, rather than strictly bounding pointwise distances. The motivation could be strengthened by either showing that this distributional view also fails or by framing MIRA as a complementary approach. I would be happy to discuss this point further, as it seems central to the paper's framing.
- **Scope of theoretical framework regarding ODE sampling**: It is unclear how MIRA extends to deterministic ODE samplers (e.g., DDIM with η=0 or modern ODE solvers), which are widely used in practice. In the deterministic case, the output is a single point, making KL divergence ill-defined (0 or ∞). Consequently, MIRA's objective may no longer serve as a direct KL surrogate. **Questions:**
	- Does the objective still act as an effective regularizer in the deterministic case? If so, what is its theoretical justification outside the stochastic KL framework?
	- This is particularly relevant for the SDXL-Turbo experiments, i.e. how should we interpret MIRA's regularization for single-step deterministic sampling?
- **Evaluation Based on Non-Standard Benchmarks:** While GPT-4o win rates are interesting, they are not a standard, reproducible benchmark for evaluating text-to-image models. The evaluation could be substantially strengthened by including results on established benchmarks like GenEval, DPG-Bench, or T2I-CompBench. This would allow for a more direct and fair comparison with the broader literature. Additionally, it would be more convincing to include quantitative results for the non-differentiable rewards instead of relying on purely qualitative. (Section 5.3)
- **Missing Implementation Details:** The main experiments in Section 5.2 do not state which reward function was used to generate the images for the GPT-4o win-rate comparison in Table 1. How many inference steps are used for SDXL-Turbo? If only one step is used (as is common), how should MIRA's multi-step regularization term be interpreted?

Minor: The GPT-4o winrate table is hard to read; some highlighting would benefit readability.

[1] Domingo-Enrich et al. "Adjoint Matching: Fine-tuning Flow and Diffusion Generative Models with Memoryless Stochastic Optimal Control". ICLR 2025.

**Questions:**

- Would using MIRA to reward fine-tune a model be equivalent to Adjoint Matching? (without the added adjoint state efficiency)
- See Weaknesses

---

> ### Author Response · Authors · 2025-11-19
> **Response to Reviewer Up1F [Part 1]**
>
> We thank the reviewer for finding our work well-motivated and for highlighting the value of our MIRA-DPO extension and the clarity of our reward hacking visualizations."
>
> > **Weakness 1, Insufficient acknowledgment of Adjoint Matching:** The paper does not discuss its relationship to Adjoint Matching [1], despite significant overlap. Adjoint Matching was proposed for reward fine-tuning with what appears to be an identical regularization approach:
>
> **Response to Weakness 1:** We thank the reviewer for this reference. While both Adjoint Matching (AM) and MIRA share the high-level goal of KL regularization, they are fundamentally different:
> 1. **Adjoint Matching Optimizes Model Weights ($\theta$):** AM is a fine-tuning algorithm. Its goal is to create a new, fine-tuned model $v^{finetune}$. The framework, including the O(1) memory adjoint-state, is designed to efficiently update the model weights.
> 2. **MIRA Optimizes Input Noise ($z$):** MIRA is a training-free, inference-time method. We explicitly freeze the backbone model. Our method optimizes the input noise $z$ via backpropagation through the frozen model without updating weights.
> 3. **The Losses are Mathematically Distinct:** The reviewer asks if our loss is equivalent to Adjoint Matching (AM). We respectfully clarify that it is not. As Equation 4 shows, our regularization term optimizes the **difference of scalar norms (magnitudes):** $$\mathcal L_\text{MIRA}\propto\sum_{t=0}^{T-1}\sigma_t^2\Big[ ||s_\text{base}(x_t)||^2-||s_\text{opt}(x_t)||^2 \Big]$$ In contrast, the regularization term in Adjoint Matching minimizes the **Euclidean distance between vectors:** $$ \mathcal L_\text{AM-reg}\propto \sum_{t=0}^{T-1}\sigma_t^2||s_\text{base}(x_t)-s_\text{opt}(x_t)||^2$$
>
> These are different objectives with distinct behaviors:
> * **MIRA (magnitude):** As long as the optimized score has the same magnitude as the base score (meaning the image is equally likely as under the unoptimized distribution), MIRA allows the vector to rotate. This flexibility is crucial for optimizing generation toward a reward.
> * **AM (distance):** By constraining the precise path, the optimized vector must match both in magnitude and direction. This penalizes deviation even if the new trajectory is equally likely, making it overly restrictive for noise optimization.
>
> **Therefore, MIRA and AM address two different problem domains.** The novelty of our work is not the derivation of a score-based regularizer in a vacuum, but its specific application and derivation for the inference-time noise optimization task which is outside the scope of AM.

---

> ### Author Response · Authors · 2025-11-19
> **Response to Reviewer Up1F [Part 2]**
>
> > **Weakness 2, Clarification on Pointwise Insufficiency:** The paper motivates its regularizer by arguing that noise-space regularization is "fundamentally insufficient" (Proposition 1), using a proof that demonstrates a pointwise property where two close noise vectors can  map to distant image distributions. This argument, however, may not fully address the core principle of existing noise-space methods, which are often viewed from a distributional perspective, aiming to keep the optimized noise $z$ within the high-probability regions of the prior $\mathcal N(0, \mathbf I)$, rather than strictly bounding pointwise distances. The motivation could be strengthened by either showing that this distributional view also fails or by framing MIRA as a complementary approach.
>
> **Response to Weakness 2:** We thank the reviewer for this distinction. We agree that methods like DNO aim to constrain the noise distribution rather than strictly bounding pointwise distance. However, we argue that **pointwise constraints are mathematically stronger** and practical optimization requires this stricter bound.
> 1. **Theoretical Justification (Pointwise implies Distributional):** Consider the KL divergence between the optimized noise distribution $p(z)$ and the prior $q(z)$. As derived in standard Gaussian KL formulations, the KL is $$\mathrm{KL}(p\|q) \approx \frac{1}{2\sigma^2}||\mu_p - \mu_q||^2+\dots$$ If we were to enforce a pointwise bound such that every optimized sample $z^p$ stays close to its initialization $z^q$ (i.e., $||z^p - z^q||\leq\epsilon$), this implicitly bounds the shift in the distribution's mean ($||\mu_p - \mu_q||\leq\epsilon$), thereby bounding the KL divergence. A pointwise constraint is therefore a **strictly stronger condition** than a distributional constraint. A distributional constraint (like DNO's) allows individual samples to drift arbitrarily far from their starting point (changing the image semantics) as long as the noise distribution remains Gaussian.
> 2. **Numerical Example:** Crucially, a distributional constraint on the noise is too loose for inference-time optimization. Consider the latent space of Stable Diffusion ($d=4 \times 64 \times 64 = 16384$) where random samples are concentrated on a shell with expected norm approximately $\sqrt d = 128$. A noise vector $z_A$ and its negation $z_B=-z_A$ have identical probability density under the prior $\mathcal N(0, \mathbf I)$. However, their Euclidean distance is $||z_A-z_B||\approx 256$.
>     * **Implication:** A distributional constraint permits shifting (by noise optimization) the noise vector across this large distance as long as the optimized noise remains on the high-probability shell. Because the mapping from noise to image is highly non-linear, such large shifts allow the image to collapse while the noise remains in the high probability region.
> 3. **Empirical Evidence:** Our experiments confirm this theoretical gap. In Figure 5's darkness example, DNO finds noise vectors that satisfy the distributional constraint (high $P(z)$) but map to degenerate black images. MIRA's image-space regularization is necessary to prevent such failure modes by constraining the denoising trajectory, not just the probability of the noise.
>
> > **Weakness 3, Scope of the theoretical framework regarding ODE sampling:** It is unclear how MIRA extends to deterministic ODE samplers (e.g., DDIM with η=0 or modern ODE solvers), which are widely used in practice. In the deterministic case, the output is a single point, making KL divergence ill-defined (0 or ∞). Consequently, MIRA's objective may no longer serve as a direct KL surrogate.
>
> **Response to Weakness 3:** We agree with the reviewer that the standard trajectory-wise KL divergence is not well-defined for deterministic paths due to the lack of absolute continuity. We have updated the manuscript to clarify that our derivation is grounded in the stochastic (SDE) formulation.
>
> **However, this does not invalidate our method for ODE samplers.** As established by Song et al. [1], "for all diffusion processes, there exists a corresponding deterministic process whose trajectories share the same marginal probability densities $\\{p_t(x)\\}_{t=0}^T$ as the SDE." Their Probability Flow ODE is constructed to match the marginal distributions $p_t(x)$ of the SDE at every timestep. Since our surrogate (Eq. 4) relies only on the score function $\nabla_x\log p_t(x)$ of these marginals rather than the path integral itself, the objective remains theoretically sound for ODE sampling such as DDIM ($\eta=0$).
>
> ---
>
> [1] Song, Yang, et al. “Score-Based Generative Modeling through Stochastic Differential Equations.” International Conference on Learning Representations, 2021.

---

> ### Author Response · Authors · 2025-11-19
> **Response to Reviewer Up1F [Part 3]**
>
> > **Weakness 4a, Evaluation Based on Non-Standard Benchmarks:** While GPT-4o win rates are interesting, they are not a standard, reproducible benchmark for evaluating text-to-image models. The evaluation could be substantially strengthened by including results on established benchmarks like GenEval, DPG-Bench, or T2I-CompBench. This would allow for a more direct and fair comparison with the broader literature.
>
> **Response to Weakness 4a:** We appreciate the suggestion to strengthen our evaluation. We utilize LLM-judged (GPT-4o) win rates because they are increasingly the standard for alignment tasks where human preference is the ground truth [2, 3, 4].
> * **Relevance to Reward Hacking:** Benchmarks like GenEval and T2I-CompBench primarily measure compositional fidelity (e.g., object counts), which is not the primary failure mode of noise optimization. Our work targets aesthetic and semantic collapse (reward hacking) which are best captured by preference-based metrics (GPT-4o or humans) rather than object detectors.
> * **Reproducibility:** To ensure reproducibility, we have provided the exact evaluation prompt in Appendix A.7.
> * **Ongoing Evaluation:** We agree, however, that standard benchmarks provide a necessary check that alignment does not degrade compositional reasoning. **We are currently running T2I-CompBench** for MIRA and will update the discussion with these results to confirm that our gains in alignment do not come at the cost of the base model's capabilities.
>
> > **Weakness 4b:** It would be more convincing to include quantitative results for the non-differentiable rewards instead of relying on purely qualitative.
>
> **Response to Weakness 4b:** This is an excellent suggestion. To address the request for quantitative evidence in the black-box setting, we conducted a head-to-head comparison using Aesthetic Score as the representative black-box objective. In Table A (added to the main paper as Table 2), we compare our MIRA-DPO against DNO's gradient approximation method (called Hybrid-2) using their reported configuration. We evaluate on the Simple Animals prompts, averaged over five matched seeds.
>
> **Table A.** Head-to-head win rate comparison with DNO's gradient approximation method. We use Aesthetic Score as a black-box reward and optimize over 50 steps on the Simple Animals dataset (45 prompts).
> | Comparison | Aesthetic Score Win Rate (%) $\uparrow$|
> | -------- | -------- |
> | MIRA-DPO vs. DNO (Hybrid-2)|$63.56\pm 3.61$|
>
> **Key Takeaway: Our MIRA-DPO outperforms DNO when treating Aesthetic Score as a black-box reward.**
>
> > **Weakness 5, Missing Implementation Details:** The main experiments in Section 5.2 do not state which reward function was used to generate the images for the GPT-4o win-rate comparison in Table 1. How many inference steps are used for SDXL-Turbo? If only one step is used (as is common), how should MIRA's multi-step regularization term be interpreted?
>
> **Response:** We thank the reviewer for the attention to detail and clarify each point as follows.
> 1. **Table 1:** We show results on three reward functions (Aesthetic, Brightness, and Darkness) which are the sub-columns for each of the three datasets (Animal, Animal-Animal, Animal-Object). We will reformat the table and headers in the final version to make this grouping much clearer and for overall readability.
> 2. **SDXL-Turbo:** We indeed use one step ($T=1$) for SDXL-Turbo and will clarify in our experimental setup. In this limit, Eq. 4 collapses to regularizing the instantaneous score norm. This constrains the magnitude of the single update vector, ensuring the noise to image mapping remains grounded in the base model's learned manifold.
>
> ---
>
> [2] Zhang, Xinlu, et al. "Gpt-4v (ision) as a generalist evaluator for vision-language tasks." arXiv preprint arXiv:2311.01361 (2023).
>
> [3] Lin, Zhiqiu, et al. "Evaluating text-to-visual generation with image-to-text generation." European Conference on Computer Vision. Cham: Springer Nature Switzerland, 2024.
>
> [4] Wu, Xun, Shaohan Huang, and Furu Wei. "Multimodal large language model is a human-aligned annotator for text-to-image generation." arXiv preprint arXiv:2404.15100 (2024).
>
> We hope these responses clarify the concerns. Please let us know if you need further clarifications.

---

> ### Author Response · Authors · 2025-11-23
> **Update regarding Standard Benchmarks**
>
> **Update regarding Weakness 4a (Standard Benchmarks):** Following up on our previous response regarding the ongoing evaluation, we have conducted T2I-CompBench experiments in the table below using PixArt-$\alpha$ (DMD 1-step) as a base model. We optimized for a composite alignment objective (ImageReward, HPSv2, PickScore, CLIPScore) to ensure comprehensive coverage of text-image alignment signals.
>
> |Method|Color|Shape|Texture|Spatial|Non-Spatial|Complex|
> |-|-|-|-|-|-|-|
> |PixArt-$\alpha$ DMD (1-step)|0.378|0.342|0.469|0.179|0.306|0.291|
> |MIRA (Ours)|**0.455**|**0.444**|**0.583**|**0.216**|**0.321**|**0.347**|
>
> **Takeaway: MIRA improves over the base model in every category.** This validates that MIRA's image-space regularization enhances compositional structure while optimizing for the target reward. We have added these results to the revised manuscript in Table 4.

---

### Author Response · Authors · 2025-11-19
**General Response Part 1: Summary of Revisions and New Evidence**

We thank all reviewers for their time and valuable feedback. We are encouraged that reviewers found our work "well-motivated" (Up1F), our "empirical results strong" (yc7h), our approach "conceptually clear" (4rBf), and our MIRA-DPO extension "valuable" (Up1F). We also appreciate Reviewer y5TU’s recognition of our "thorough" mechanism analysis.

We have uploaded a revised manuscript (highlighted as blue text) incorporating the following key updates based on your constructive suggestions:

1. **New Theoretical Validation:** To address requests for rigorous validation of our score-based surrogate (Reviewer y5TU), we have added a new Figure 5 to the Appendix. This experiment confirms a strict monotonic correlation between our MIRA surrogate and the true analytical KL divergence in a controlled setting.

2. **Quantitative MIRA-DPO Evaluation:** To address the request for quantitative results on non-differentiable rewards (Reviewer Up1F), we added Table 2 in the main text, demonstrating a 63.56% win rate for MIRA-DPO against the DNO baseline on Aesthetic Score as a black-box reward.

3. **Visualization Consistency:** We have regenerated Figures 1 and 5 in the main text and Figure 1 in the appendix with strictly fixed seeds to ensure a clean comparison of semantic drift, as suggested by Reviewer yc7h.

4. **Theoretical Convergence with Concurrent Work:** We note that our core theoretical premise that noise-space constraints are insufficient for semantic preservation has been independently validated by a very recent preprint (Li & He, [1]). This concurrent finding corroborates our methodological shift from noise-space to image-space score regularization.

We have also posted individual responses addressing each reviewer's specific concerns. We look forward to the discussion.

---

[1] Li, Tianhong and Kaiming He. "Back to Basics: Let Denoising Generative Models Denoise." arXiv 2025.

---

### Author Response · Authors · 2025-11-23
**General Response Part 2: New Experimental Results (T2I-CompBench and DiT)**

To address requests for DiT architectures (Reviewer y5TU) and standardized benchmarks (Reviewer Up1F), we extended MIRA to PixArt-$\alpha$ (DMD 1-step) and evaluated it on T2I-CompBench as shown in the table below.

**Table:** T2I-CompBench Results on PixArt-$\alpha$. We optimized a standard composite objective (ImageReward, HPSv2, PickScore, CLIPScore) following the formulation in recent literature to ensure a robust alignment signal.

|Method|Color|Shape|Texture|Spatial|Non-Spatial|Complex|
|-|-|-|-|-|-|-|
|PixArt-$\alpha$ DMD (1-step)|0.378|0.342|0.469|0.179|0.306|0.291|
|MIRA (Ours)|**0.455**|**0.444**|**0.583**|**0.216**|**0.321**|**0.347**|

**Takeaways:** MIRA consistently improves over the base model (e.g., +30% on Shape, +24% on Texture, +19% on Complex). This demonstrates that MIRA's image-space regularization generalizes effectively to diffusion transformers and enhances compositional alignment on standard benchmarks.

---

> ### Comment · Reviewer_y5TU · 2025-11-25
>
> Applying an alignment method to some specific tasks and gain improvements is undoubted, which means the baseline shouldn't be PixArt itself. Authors should compare some meaningful baselines like DNO, PRNO, etc.

---

> ### Author Response · Authors · 2025-11-25
> **Response regarding Baseline Comparison**
>
> Thank you for the feedback. We agree that comparing against optimization baselines is important to contextualize performance.
>
> As requested, we compared MIRA against ReNO, a noise optimization baseline, on PixArt-$\alpha$ using identical seeds. We find that **MIRA significantly outperforms ReNO on the Complex category (0.347 vs 0.308), a relative improvement of +12.7%.**
>
> **Takeaway:** "Complex" is the most challenging subset of T2I-CompBench, requiring the model to bind multiple attributes and spatial relationships **simultaneously.** MIRA's improvement here suggests that our image-space regularization preserves global structural coherence more effectively than noise-space constraints.
>
> Detailed comparisons with regularized DNO (i.e., PRNO) and ReNO across multiple rewards and datasets are provided in the main paper (Section 5.2), demonstrating that **MIRA consistently outperforms these baselines in win-rate comparisons (often exceeding 60%)**. This experiment serves to validate that our structural preservation benefits generalize to DiT architectures.

---

### Comment · Area_Chair_atUa · 2025-11-27
**Please check the rebuttal**

Dear Reviewers,

The authors have posted their rebuttal. Could you please take a moment to review their responses and check whether your concerns have been adequately addressed? If possible, kindly initiate the discussion at your earliest convenience.

Your timely assistance is essential for keeping the review process on track. Thank you very much for your support and contribution.

Best regards,
Your AC

---

### Author Response · Authors · 2025-11-28
**Summary of Rebuttal and Key Revisions (For New Area Chair)**

Dear New Area Chair,

In light of the administrative updates, we provide a summary of how we resolved the primary concerns raised in the original reviews. We addressed all distinct technical concerns with new experiments and theoretical clarifications:
1. Addressed "Standard Benchmarks and Quantitative Eval" (Up1F)
    * We evaluated MIRA on **T2I-CompBench**. MIRA consistently improves over the base model on all metrics (e.g., **+30% on Shape**), demonstrating robust alignment. (see **General Response Part 2**).
    * We added Table 2 (main text) comparing MIRA-DPO against DNO for **black-box** rewards, demonstrating a **63.5% win rate**, validating our inference-time DPO formulation.
2. Addressed "Scope and Generalization" (y5TU, yc7h)
    *  We extended MIRA to a **Diffusion Transformer (DiT)** architecture (PixArt-$\alpha$), validating that our method generalizes beyond UNets. (See **General Response Part 2**).
3. Addressed "Baseline Comparisons" (y5TU, 4rBf)
    *  We compared against **ReNO** (state-of-the-art) on PixArt using identical seeds. MIRA significantly outperformed ReNO on the **"Complex"** benchmark (**+12.7%**), validating our claim of structural integrity (see **Response to Reviewer y5TU**).
4. Addressed "Theoretical Novelty and Validity" (Up1F, y5TU)
     * Adjoint Matching: We clarified the fundamental mathematical distinction (optimizing score magnitude difference vs. vector Euclidean distance) between our work and Adjoint Matching (see **Response to Up1F**).
     * Theory: We added a 1D Toy Experiment (Figure 5, Appendix) confirming the monotonic correlation of our surrogate with the true KL, and clarified the ODE/SDE formulation (see **General Response Part 1**).

5. Addressed "Efficiency and Visuals" (4rBf, yc7h)
    * We clarified that MIRA steps are time-equivalent to DNO steps (backprop bottleneck) and regenerated all qualitative figures with fixed seeds to ensure rigorous comparison.

Overall, during the discussion, Reviewer y5TU acknowledged the **"clear contribution towards tackling the reward hacking problem."** Reviewers Up1F and yc7h highlighted the work as **"well-motivated"** and having **"strong empirical results."** Reviewer Up1F praised MIRA-DPO as a **"valuable contribution"** extending applicability.

---

### Meta-Review · Area_Chair_mFKP · 2026-01-09

**Summary:**

The paper proposes an approach for improving inference-time alignment of generative models with reward models, dubbed MIRA. The approach is to add a regularizer along the sampling trajectory of the model. In addition, the authors propose a version of the approach, MIRA-DPO, applicable to non-differentiable rewards. The method is evaluated with several reward models and generative models and shows favorable performance compared to baselines.

Based on the reviews, authors’ rebuttal, and the paper itself, here are the key pros and cons.

Pros:
1. For multi-step models, good performance, especially for SD1.5, at little/no extra cost compared to DNO
2. Also improved performance for one-step models compared to ReNO.
3. The authors addressed some reviewers’ concerns in the rebuttal - added T2I-CompBench results, DiT results, comparison to ReNO, and some more clarifications and improvements.

Cons:
1. Not fully conclusive results: A large advantage over DNO with SD1.5, medium vs ReNO with SDXL-Turbo, small one for SDXL.
2. Relation to adjoin matching not discussed in the paper. It has been discussed in the rebuttal, but I don’t see it in the paper at the moment. Moreover, the distinction between difference of norms and difference in vectors, as pointed out in the rebuttal, makes sense, but the explanation in the rebuttal of how regularizing the difference of the vectors is too constrained is hand-wavy. For this argument to be fully convincing, it would have to be supported with experiments. (Also, the point that Adjoint matching is training time and MIRA inference-time is clear, and the reviewer referred to it too - their point was that the same regularization can be used at training and inference time)
3. In Table 4 for DiT, the comparison seems to be against the raw model, not against a competing inference-time method. This is not very informative - I mean, inference-time method being better than the model itself is a necessary condition for it to be useful, but not a sufficient condition for the contribution to be meaningful.
4. My understanding is that Table 2 compares the methods based on reward value, and a higher reward is better. If so, this seems counter to the paper’s narrative about reward hacking. I mean, again, optimizing reward at least to some degree is likely a necessary condition for an inference-time alignment approach to work, but the assumption that higher is better seems generally wrong.
5. On a related topic, I find Figure 4 a bit confusing in that the reward of MIRA does not go up at all. I guess it’s because the first point is not at 0 steps, and it does go up between 0 and this first point? Moreover, how can one understand in this figure when rewards going up is ok and real, and when it’s hacking? Additionally, Figure 6 right, is a bit confusing in that there the rewards do go up.

Overall, while the paper has its merit, there are a few issues, as the ones listed in Cons above, that at the moment do not let the paper be accepted as-is. The contribution is non-trivial, but quite related to other works (other inference-time methods, adjoint matching, etc), so it has to be executed really well and perform really well for it to be accepted. As of now, the work does show a fair bit of good performance and interesting analysis, but does not quite meet the bar.

**Reviewer Concerns:**

- Standard evaluations, not just comparison by VLMs. -> The authors added T2I-CompBench results (not super convincing though, only vs a base model without inference-time alignment)
- Quantitative evaluation of MIRA-DPO -> Added some quantitative evaluation (seems not sure convincing though, see above)
- More architectures/models -> the authors added DiT results (again, not very convincing since compared to a base model)
- More baseline comparisons -> added a comparison to ReNO. Looks pretty good.
- Relation to adjoint matching -> discussed in the rebuttal, but not in the paper
- Computational efficiency / timing -> clarified that it’s basically the same as DNO

**Reviewer Scores:**

I think 1-2 reviewers might increase the scores by 1 point since some concerns have been addressed.

---

### Decision · Program_Chairs · 2026-01-26

Reject